DOI: 10.1038/s41467-018-02891-z · OPEN

# Analysis of cardiomyocyte clonal expansion during mouse heart development and injury

Konstantina-Ioanna Sereti[1,2], Ngoc B. Nguyen[1,2,3], Paniz Kamran[1,2], Peng Zhao[1,2], Sara Ranjbarvaziri[1,2,3], Shuin Park[1,2,3], Shan Sabri[2,4,5,6], James L. Engel[1,2,3], Kevin Sung[7], Rajan P. Kulkarni[5,7], Yichen Ding[1], Tzung K. Hsiai[1], Kathrin Plath[2,4,5,6], Jason Ernst[2,4,5,6], Debashis Sahoo[8], Hanna K.A. Mikkola[2,5,9], M. Luisa Iruela-Arispe[2,9,10] & Reza Ardehali[1,2,3,5,10]

The cellular mechanisms driving cardiac tissue formation remain poorly understood, largely due to the structural and functional complexity of the heart. It is unclear whether newly generated myocytes originate from cardiac stem/progenitor cells or from pre-existing cardiomyocytes that re-enter the cell cycle. Here, we identify the source of new cardiomyocytes during mouse development and after injury. Our findings suggest that cardiac progenitors maintain proliferative potential and are the main source of cardiomyocytes during development; however, the onset of αMHC expression leads to reduced cycling capacity. Single-cell RNA sequencing reveals a proliferative, "progenitor-like" population abundant in early embryonic stages that decreases to minimal levels postnatally. Furthermore, cardiac injury by ligation of the left anterior descending artery was found to activate cardiomyocyte proliferation in neonatal but not adult mice. Our data suggest that clonal dominance of differentiating progenitors mediates cardiac development, while a distinct subpopulation of cardiomyocytes may have the potential for limited proliferation during late embryonic development and shortly after birth.

[1] Division of Cardiology, Department of Internal Medicine, David Geffen School of Medicine, University of California, Los Angeles, Los Angeles, CA 90095, USA. [2] Eli and Edythe Broad Center for Regenerative Medicine and Stem Cell Research, University of California, Los Angeles, Los Angeles, CA 90095, USA. [3] Molecular, Cellular and Integrative Physiology Graduate Program, University of California, Los Angeles, Los Angeles, CA 90095, USA. [4] Department of Biological Chemistry, University of California, Los Angeles, Los Angeles, CA 90095, USA. [5] Jonsson Comprehensive Cancer Center, Los Angeles, CA 90095, USA. [6] UCLA Bioinformatics Interdepartmental Program, University of California, Los Angeles, Los Angeles, CA 90095, USA. [7] Division of Dermatology, Department of Medicine, David Geffen School of Medicine, University of California, Los Angeles, Los Angeles, CA 90095, USA. [8] Departments of Pediatrics and Computer Science and Engineering, University of California San Diego, La Jolla, CA 92093, USA. [9] Department of Molecular, Cell and Developmental Biology, University of California, Los Angeles, Los Angeles, CA 90095, USA. [10] Molecular Biology Institute, University of California, Los Angeles, Los Angeles, CA 90095, USA. Konstantina-Ioanna Sereti, Ngoc B. Nguyen, Paniz Kamran contributed equally to this work. Correspondence and requests for materials should be addressed to R.A. (email: RArdehali@mednet.ucla.edu)

The adult mammalian heart has long been considered a non-regenerative organ and cardiomyocytes (CMs), the building blocks of the heart, as terminally differentiated cells. A number of studies have demonstrated a low rate of CM turnover[1–3] while others have suggested the existence of distinct CM populations that maintain their proliferative capacity throughout adulthood[4]. Remarkably, zebrafish[5] as well as neonatal mice[5,6] can efficiently regenerate their hearts in response to injury. A recent study by Sturzu et al.[7] reported the ability of the embryonic heart to rapidly restore extensive tissue loss through robust CM proliferation. However, the proliferative capacity of CMs during development and after birth remains an area of controversy. It is unclear whether newly generated myocytes originate from cardiac stem/progenitor cells or from pre-existing CMs that re-enter the cell cycle. In this paper, we utilized the Rainbow system to perform clonal analysis of CMs during development and after injury to obtain a better mechanistic understanding of cardiac growth. The Rainbow system marks a small number of cells and their progeny with a distinct fluorescent protein, allowing retrospective tracing of cellular expansion through easily identifiable clones in vivo.

Through single-cell lineage tracing, we find that cardiomyocytes marked as early as embryonic day 9.5 (E9.5) have the capacity to form large clones both in vitro and in vivo; however, this capacity is substantially reduced by E12.5. Additionally, our data suggest the possibility that cardiovascular progenitors contribute to the majority of cardiac growth during embryonic development and that their maturation occurs with gradual expression of cardiac-specific markers concomitant with their decreasing proliferative capacity. Single-cell RNA sequencing supports the notion of heterogeneity in the proliferative capacity of αMHC-expressing CMs over time. Within the early stages of cardiac development, we observe a potential reduction in developmental growth signals and a shift toward pathways involved in heart contraction and cellular respiration. Taken together, our study provides important insights into the source of CMs and the characteristics of progenitor cells both during development and after injury.

## Results

**Rainbow provides a direct tool for clonal expansion analyses.** To study clonal distribution in the heart, we used Rainbow (hereafter termed $R26^{VT2/GK}$) mice (Supplementary Figure 1a) to generate transgenic lines expressing Cre under the control of early cardiovascular progenitor transcription factors, Mesp1[8] and Nkx2.5[9]. In this system, Cre-mediated recombination of paired lox P sites results in permanent expression of a distinct fluorescent protein in cells expressing Mesp1 or Nkx2.5 and their subsequent progeny (Fig. 1a–f). There was equal expression of all three fluorescent proteins (mCherry, mOrange, and Cerulean) in the labeled hearts (Supplementary Figure 1b–c). We analyzed heart sections at embryonic day 14.5 (E14.5), and postnatal days 1 and 21 (P1, P21) for the presence of clonal patterns. We identified single-color multicellular clusters evident across the atria and ventricles, indicating possible clonal areas (Fig. 1b, e; Supplementary Movie 1). We expected some uniformity to the clonal regions in terms of shape and size; however, hearts from different animals displayed unique clonal distribution patterns. To examine the proliferative behavior of CMs during development, we utilized $αMHC^{Cre};R26^{VT2/GK}$ mice (Fig. 1g–i)[10]. αMHC-derived

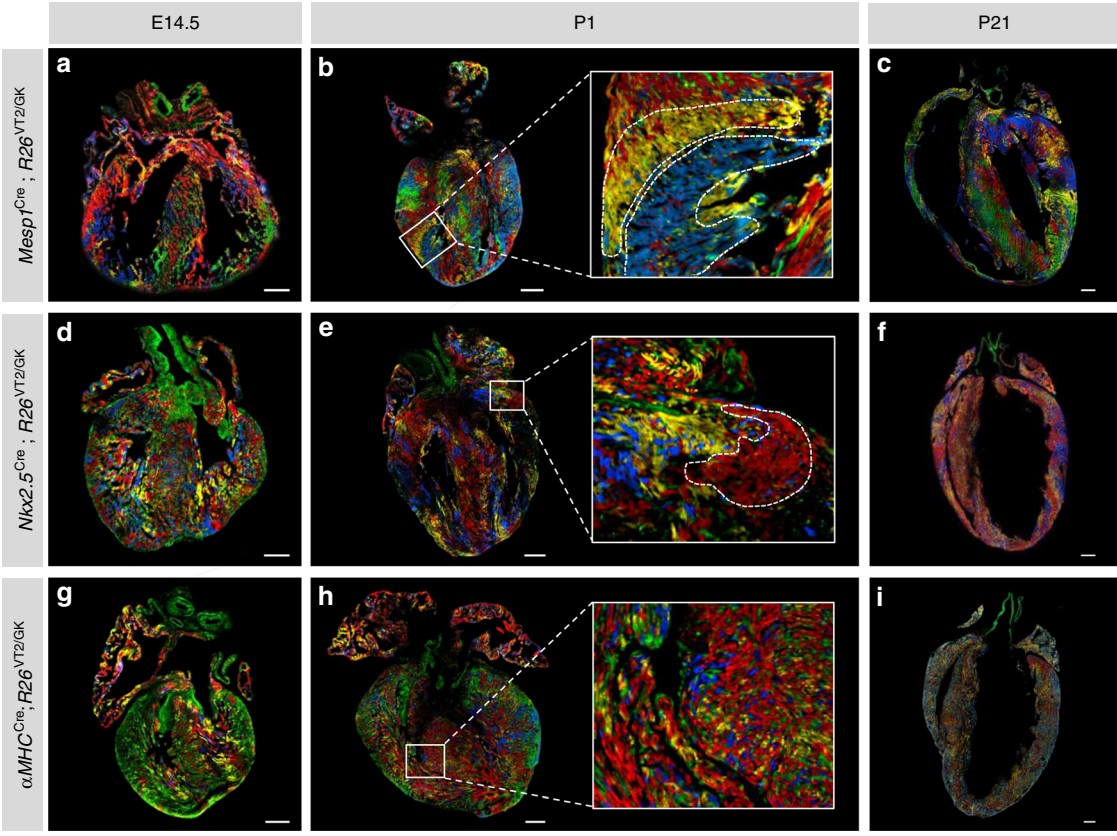

**Fig. 1** Evidence of clonal expansion during cardiac development. Representative fluorescent microscope images of **a–c**, $Mesp1^{Cre}; R26^{VT2/GK}$, **d–f**, $Nkx2.5^{Cre}; R26^{VT2/GK}$, and **g–i**, $αMHC^{Cre}; R26^{VT2/GK}$ longitudinal hearts sections at E14.5, P1, and P21 respectively ($n = 4$). Insets show a higher magnification of boxed areas. Putative clonal areas are traced by a dotted line. Scale bar 500 μm (**c**, **f**, **i**), all others 50 μm

CMs appeared mainly as singletons, illustrated by the mosaic appearance of Rainbow colors and the limited occurrence of single-colored clones (Fig. 1h inset). These data suggest that cardiac progenitors marked by the expression of Mesp1 and Nkx2.5 are potential sources of substantial CM expansion, whereas αMHC-expressing CMs possess low-cycling capacity, even during embryonic development.

**Single-cell lineage tracing of early cardiac progenitors**. However, the use of Rainbow with a non-inducible Cre model results in high levels of recombination. To exclude the possibility that the observed single-color cell clusters could result from random recombination and expression of the same fluorescent protein within neighboring cells, we used tamoxifen-inducible Cre lines that permit tight spatiotemporal control on recombination events (Fig. 2a–f). We initially investigated the clonal expansion of all cell types in the heart (including CMs and stem/progenitor cells) by crossing $R26^{VT2/GK}$ mice with mice harboring an inducible Cre under the control of a β-actin ($\beta actin^{CreER}$) promoter (Fig. 2a, d). Since β-actin is ubiquitously expressed, fluorescent labeling could occur in any cell. We achieved single-cell labeling through titration of tamoxifen dosage (Supplementary Figure 2a), which generated easily identifiable clones for analysis (Supplementary Figure 2b). Twenty-four hours following administration, serum tamoxifen level was reduced to negligible levels, ensuring tight temporal control of recombination (Supplementary Figure 2c). In addition, there were no significant differences in the distribution or frequency of the three fluorescent proteins in single-cell labeling and accordingly, there was comparable prevalence of the different colored clones at the time of analysis.

Within $\beta actin^{CreER}$; $R26^{VT2/GK}$ mouse hearts, we observed fibroblast, endothelial, and smooth muscle cell clones (Supplementary Figure 3a–d); however, we focused our analysis on CM clusters. When recombination was induced at E9.5, we identified single-colored clones ranging from 2 to 40 cells, with an average of 5.7 cells/clone at all timepoints analyzed (Fig. 2a and Supplementary Figure 4a–b). Clone size quantification revealed slightly larger clones at P7, P15, and P30 compared to P2 (Supplementary Figure 4b), which agrees with several studies suggesting limited proliferative capacity of CMs during early postnatal life[11–13]. Interestingly, we observed a drop in the number of clones larger than 20 cells at P30 (Supplementary Figure 4b), which was accompanied by an increase in single "Rainbow-colored" cells (Supplementary Figure 4c). Possible explanations for the higher incidence of single cells include migration of daughter cells after division within a clone or dispersion of sibling cells in a clone among other dividing neighboring cells. Isolation and in vitro culture of single "Rainbow-labeled" cardiac cells from E9.5 $\beta actin^{CreER}$; $R26^{VT2/GK}$ embryos showed that these cells retained their ability to form clones through several rounds of division (Supplementary Figure 5). These findings suggest that cardiac cells marked as early as E9.5 have the capacity to form large clones both in vivo and in vitro.

To determine whether cardiac progenitors are the main contributors to the observed CM clonal expansion, we generated an inducible Nkx2.5 reporter mouse ($Nkx2.5^{CreER}$) by targeting an IRES-CreER sequence to the 3′ end of the endogenous Nkx2.5 locus, in order to avoid haploinsufficiency (Supplementary Figure 6). Clonal analysis of $Nkx2.5^{CreER}$; $R26^{VT2/GK}$ mice labeled at E9.5 and analyzed at P2 revealed a similar pattern of clonal expansion, with the largest clone containing 39 cells, and an average of 7.5 cells/clone (Figs. 2b and 3a). Collectively, these data support the proliferative capacity of progenitor cells to generate myocardial cells during early fetal development.

**αMHC-marked cells decrease their proliferation over time**. Although these studies support the active division of cardiac progenitors during early cardiovascular development, we sought to determine the proliferative behavior of CMs at similar stages of development. To accomplish this, we crossed $R26^{VT2/GK}$ with mice harboring an inducible Cre under the control of an αMHC ($\alpha MHC^{CreER}$) promoter. Expression of αMHC in embryonic mouse hearts was observed at E9.5, and by E12.5, most of the distribution was found in the atria and to a lesser extent in the ventricles (Supplementary Figure 8b). Analysis of $\alpha MHC^{CreER}$; $R26^{VT2/GK}$ mice labeled at E9.5 revealed smaller clones (largest observed consisted of 30 cells, with an average of 6.3 cells/clone) compared to $\beta actin^{CreER}$; $R26^{VT2/GK}$ and $Nkx2.5^{CreER}$; $R26^{VT2/GK}$ in which clones >50 cells were observed (Figs 2c and 3a). Arbitrary clone size cutoff (dashed lined) was determined as two standard deviations above the clone size mean of $\alpha MHC^{CreER}$; $R26^{VT2/GK}$. Comparison of the percent of clones within each strain larger than this cutoff showed similar percentages at E9.5 (Fig. 3b). Interestingly, the majority of clones were localized to the inner endocardial region (Supplementary Figure 7a) and phospho-H3 staining in E9.5 hearts was mainly localized to the myocardial compact zone, suggesting that clonal growth may be directed towards the endocardium (Supplementary Figure 7b). Furthermore, β-actin- and Nkx2.5-expressing cells labeled at E12.5 exhibited a slight decrease in proliferation compared to at E9.5 (up to 26 cells/clone in β-actin, and 21 cells/clone in Nkx2.5) (Figs. 2d, e and 3c). However, a dramatic decrease was observed in αMHC cells labeled at E12.5, with the largest clone consisting of only nine CMs when analyzed postnatally. Additionally, the percentage of clones larger than two cells decreased by 30% compared to labeling at E9.5, with exceedingly rare clones consisting of more than five CMs (Figs. 2f and 3d). This suggests that αMHC-expressing CMs at E9.5 retain the ability to proliferate (albeit to a lesser degree than Nkx2.5 cardiovascular progenitors), and that this capacity is significantly diminished by E12.5.

Our two dimensional (2D) quantification revealed striking differences in clone sizes that emerged from labeling single progenitors or CMs at different time points. However, there is possibility for error in clone size estimation, as the depth of each cell cluster cannot be accounted for using cross sections. To address this, we developed a modified CLARITY technique[14] to transform the intact heart into an optically transparent but structurally preserved organ for light sheet fluorescence microscopy (Supplementary Figure 9a–b). This methodology facilitates three-dimensional (3D) imaging of whole-neonatal mouse hearts to precisely quantify and compare clone volumes from each strain at various time points (Supplementary Figure 9c, Supplementary Movie 2). To provide a parallel comparison with our 2D results, we labeled the three strains at both E9.5 and E12.5. Analysis of P2 cleared hearts from $\beta actin^{CreER}$; $R26^{VT2/GK}$ and $Nkx2.5^{CreER}$; $R26^{VT2/GK}$ mice labeled at E9.5 revealed clones of similar volumes (average $190,220\,\mu m^3$ and $214,260\,\mu m^3$, respectively), which decreased by approximately twofold when labeling occurred at E12.5 (Fig. 3e, g). In $\alpha MHC^{CreER}$; $R26^{VT2/GK}$ hearts labeled at E9.5 and analyzed at P2, we did not find a difference in clone volumes compared to $Nkx2.5^{CreER}$; $R26^{VT2/GK}$ and the percent of clones within each strain with volumes above $100,000\,\mu m^3$ were similar (Fig. 3f). However, when labeling was initiated at E12.5, significant differences were observed in volumes of αMHC-marked clones compared to β-actin and Nkx2.5 ($p < 0.001$) and there was a substantial decrease in the percent of clones within $\alpha MHC^{CreER}$; $R26^{VT2/GK}$ that were above the volume cutoff compared to $Nkx2.5^{CreER}$; $R26^{VT2/GK}$ (Fig. 3h). Additionally, within aMHC-marked clones, there was a ~16-fold decrease in volumes with labeling at E12.5 when compared to E9.5. These results further validate the notion that while αMHC-

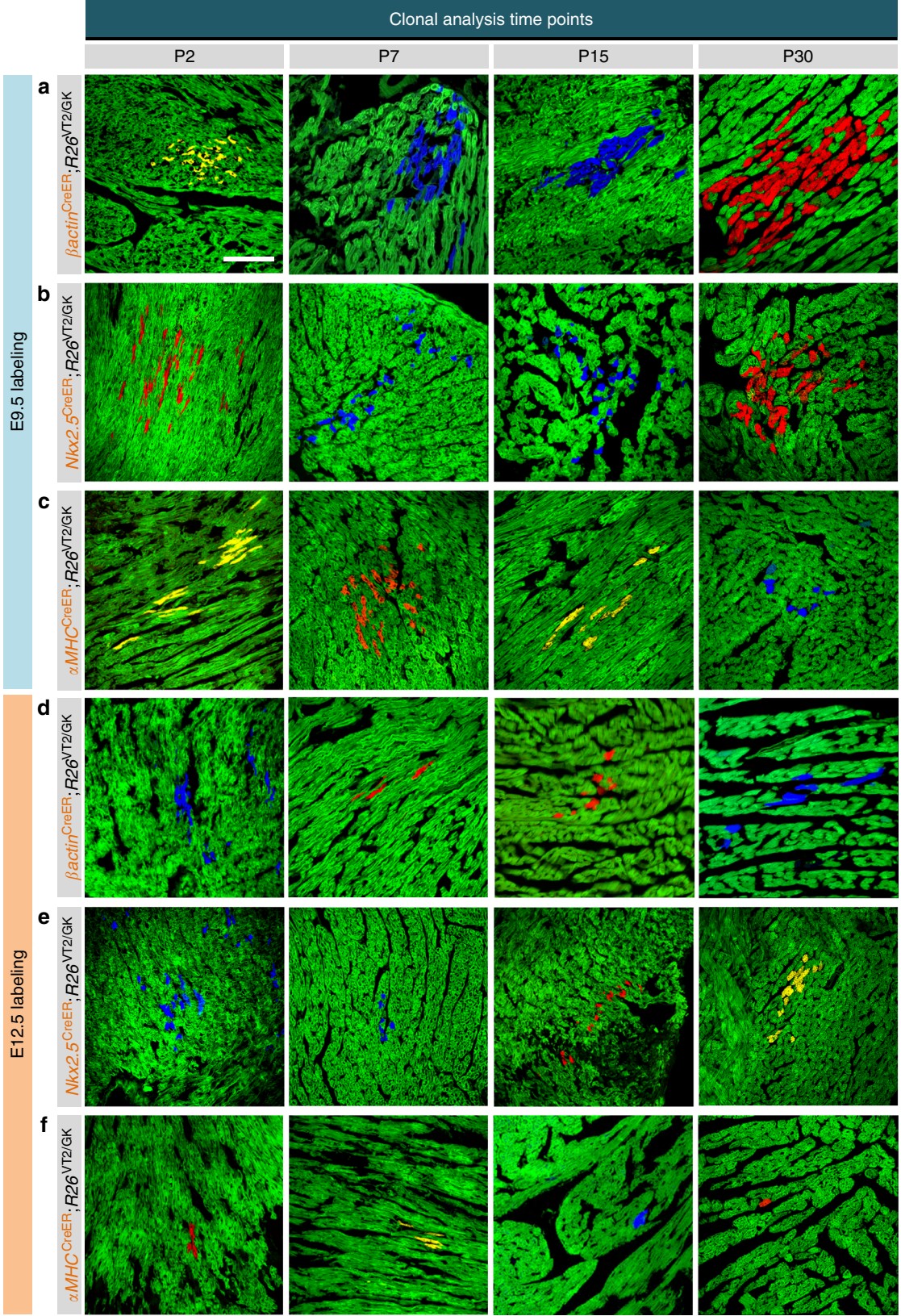

**Fig. 2** Cardiac growth occurs primarily through clonal expansion of non-αMHC-expressing cells. **a** Representative confocal microscope images of cell cluster expansion in **a,d** $\beta actin^{CreER}$; $R26^{VT2/GK}$, **b,e** $Nkx2.5^{CreER}$; $R26^{VT2/GK}$ and **c,f** $\alpha MHC^{CreER}$; $R26^{VT2/GK}$ hearts at different developmental timepoints, following tamoxifen administration at E9.5 (**a–c**) and E12.5 (**d–f**) ($n = 6$ for each timepoint). Scale bar 100 μm

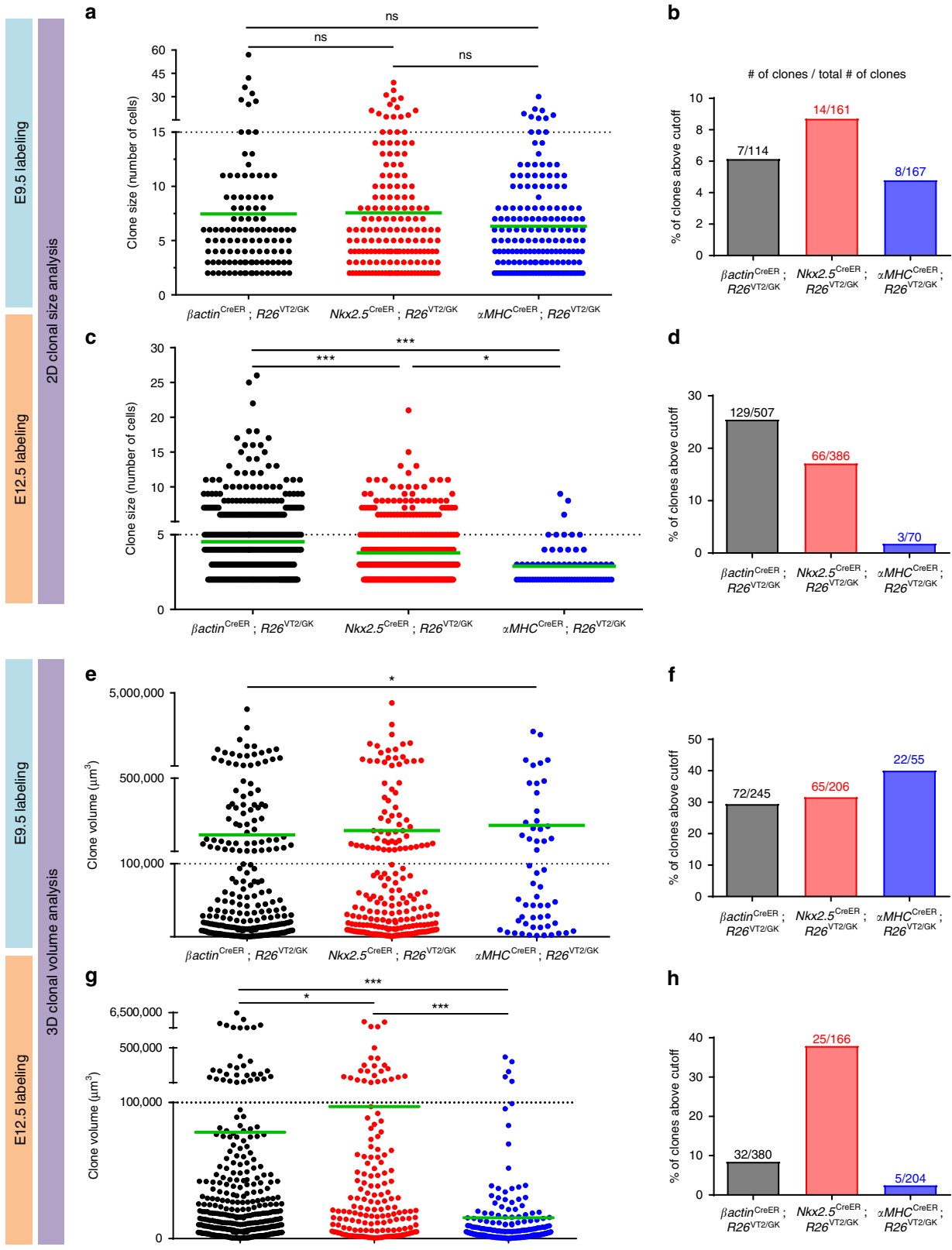

**Fig. 3** 2D and 3D quantification reveals a dramatic decrease in the proliferative capacity of αMHC-expressing cells. Quantification of clonal expansion in P2 $\beta actin^{CreER}$; $R26^{VT2/GK}$, $Nkx2.5^{CreER}$; $R26^{VT2/GK}$, and $\alpha MHC^{CreER}$; $R26^{VT2/GK}$ hearts labeled at E9.5 (**a**, **b**, **e**, **f**) and E12.5 (**c**, **d**, **g**, **h**) (two-sample Kolmogorov–Smirnov distribution test) *$p < 0.05$, **$p < 0.01$, ***$p < 0.001$. Two dimensional quantification of clonal expansion (**a**, **c**) and bar graphs highlighting the scarcity of large clones in $\alpha MHC^{CreER}$; $R26^{VT2/GK}$ hearts (**b**, **d**), respectively. Three-dimensional quantification of clone volumes in P2 hearts (**e**, **g**) and bar graphs depicting clones volumes in $\alpha MHC^{CreER}$; $R26^{VT2/GK}$ hearts (**f**, **h**). Green lines depict mean values. Dashed lines depict cutoff

marked CMs retain the ability to proliferate during early embryonic development, cardiovascular progenitors are the primary contributors of cardiac growth during this time.

To exclude the possibility that lack of clone formation in $\alpha MHC^{CreER}$; $R26^{VT2/GK}$ hearts is due to the inability of the system to specifically label CMs with proliferative potential, we stimulated CM proliferation with neuregulin (NRG1), an extracellular growth factor shown to promote CM cell cycle re-entry in normal and injury conditions[15] (Supplementary Figure 10). $\alpha MHC^{CreER}$; $R26^{VT2/GK}$ mice received tamoxifen to induce sparse labeling of CMs at P15 followed by daily administrations of NRG1 from P20 to P28. Scoring the frequency of clones revealed an increase in "Rainbow-labeled" CM clones in NRG1 treated hearts compared to control (fourfold increase in doublets) (Supplementary Figure 10b–c). This validates that the decreased clonal size observed in $\alpha MHC^{CreER}$; $R26^{VT2/GK}$ hearts during development is due to the limited intrinsic ability of these cells to divide and further support the validity of the Rainbow system as a sensitive tool for clonal analysis.

Our results thus far suggest that cardiac development is a continuum in which progenitors progressively transition to more mature, less proliferative CMs. Interestingly, a recent prior study[16] reported a burst of highly synchronized CM proliferation during preadolescence. This proliferative event, occurring at P14, resulted in a remarkable 40% increase in CM number. To examine whether the decrease in CM proliferation we observed during embryonic development is followed by a re-activation of cell division postnatally, we labeled single CMs at P12 using $\alpha MHC^{CreER}$; $R26^{VT2/GK}$ mice and looked at their proliferative potential retrospectively at P30. Consistent with other reports[12,17], we observed mainly single-labeled cells dispersed throughout the myocardium and rare clones of 2–3 cells (Supplementary Figure 8a). Furthermore, we detected rare clusters of CMs consisting of >2 cells derived from β-actin-, Nkx2.5-, or αMHC-Rainbow labeling at P1, suggesting limited expansion of CMs even prior to preadolescence (Supplementary Figure 8c). Collectively, these findings do not support the substantial second wave of CM proliferation during preadolescence as was described.

**Epicardial progenitors are unlikely to be sources of CMs.** Although our study demonstrates the source of CMs during development, it does not address whether stem/progenitor cells are entirely endogenous to the myocardium. There is controversy regarding the contribution of epicardial cells to the developing heart. Several studies have reported that epicardial cells can differentiate into endothelial cells, vascular smooth muscle cells, fibroblasts, and even CMs[18,19]. To examine whether the observed clones may have originated from epicardial-derived progenitor cells, rather than intramyocardial progenitors or CMs, we utilized a $Wt1^{CreER}$; $R26^{VT2/GK}$ double-transgenic mouse (Supplementary Figure 11a). Postnatal analysis of cells labeled at E9.5 or E12.5 did not reveal any Rainbow-labeled CMs, but the clones identified primarily consisted of fibroblasts and in some cases, smooth muscle cells (Supplementary Figure 11b–c).

**BrdU studies uncover decreased proliferative capacity of CMs.** To further support our clonal analysis data, in vivo BrdU pulse/ chase experiments were performed. BrdU was administered to dames carrying $\alpha MHC$-GFP embryos at E9.5 or E12.5 and to P1 neonates 3 h prior to heart harvest. Flow cytometric analysis of αMHC+ cells revealed a dramatic decrease in the percentage of BrdU+ CMs from E9.5 to E12.5 (~ninefold decrease) and P1 (~60-fold decrease) (Fig. 4a, b and Supplementary Figure 12a). We next evaluated the proliferation of αMHC-expressing CMs

relative to cardiac progenitors by performing a similar pulse/ chase experiment in triple transgenic mice ($Nkx2.5^{+/Cre}$; $R26R$-$Tdt^{+/fl}$; $\alpha MHC^{+/GFP}$). In this model, cardiac progenitors ($Nkx2.5^+/\alpha MHC^-$) express only tdTomato while αMHC-expressing CMs ($Nkx2.5^+/\alpha MHC^+$) express both tdTomato and GFP. We observed that while the fraction of αMHC-expressing CMs increased from E9.5 to E12.5 and P1 (Supplementary Figure 12a), their proliferative capacity at E12.5 and P1 decreased by ~sevenfold and 65-fold compared to E9.5, respectively, as indicated by the decrease in percentage of BrdU incorporation (Fig. 4c). Importantly, we observed a significantly lower proportion of BrdU+ cells at E12.5 and P1 within the CM population compared to progenitors even though their BrdU incorporation was relatively similar at E9.5 (Fig. 4d). Correspondingly, expression of cell cycle markers (Ccnb, Cdc6, and Ccna) in GFP + CMs isolated from αMHC-GFP mice were higher at E9.5 compared to later time points (Fig. 4e), and this was inversely correlated with αMHC expression levels (Fig. 4f). These data suggest that as the embryonic heart develops, αMHC-expressing cells become progressively more committed, while progenitor cells retain their proliferative potential for a longer span of time. It is possible that αMHC marks a heterogeneous population of CMs that differ in their proliferative capacity and maturity level; less mature αMHC-expressing cells may exhibit higher proliferative potential, whereas more mature αMHC-expressing CMs (found in abundance at E12.5 and beyond) are limited in their ability to undergo division. We therefore hypothesized that heart formation is a dynamic process that consists of CMs with varying proliferative potential and that these populations are refined as development proceeds.

**Single-cell RNA sequencing delineates heterogeneity of CMs.** To test this hypothesis and to profile the potential heterogeneity of CMs during early cardiac development, we used a Fluidigm C1 chip platform to capture αMHC+ cells from αMHC-GFP murine hearts at E9.5, E12.5, and P1 for single-cell RNA sequencing. Overall, we analyzed a total of 122 cells that passed quality screening. Unsupervised dimensionality reduction by t-SNE identified clusters of single cells that appeared to correspond to their developmental time point (Fig. 5a). Heat map analysis of these cells on genes relevant to cardiac development and maturation showed that P1 CMs displayed a more mature, less proliferative transcriptional profile that is distinguishable from E9.5 and E12.5 clusters (Fig. 5b). In particular, genes encoding for cell cycle, cardiac cell differentiation, and cellular migration were downregulated in P1 cells while these same markers were expressed at high levels in the majority of E9.5 and E12.5 CMs. Conversely, the expression of genes encoding for structural proteins and cellular metabolism were upregulated in the P1 population but expressed at low levels in the earlier time points. We also observed substantial heterogeneity in gene expression levels within E9.5 and E12.5 cells compared to P1. Expression of cell cycle-associated genes (Ccnb1, Cdc6, and Ccna2) from individual cells at each time point (Fig. 5c) confirmed our qPCR results showing a decline in cell cycle activity from E9.5 to P1 (Fig. 4e). Additionally, consistent with our hypothesis that early embryonic CMs span a broad spectrum in their capacity to proliferate, we observed a wide range of gene expression levels of cell cycle markers at E9.5 that progressively narrowed at later time points. These results provide initial evidence to support the existence of a heterogeneous population of CMs within the early stages of cardiac development and their transition into a mature, less proliferative, and homogenous population by the early postnatal period.

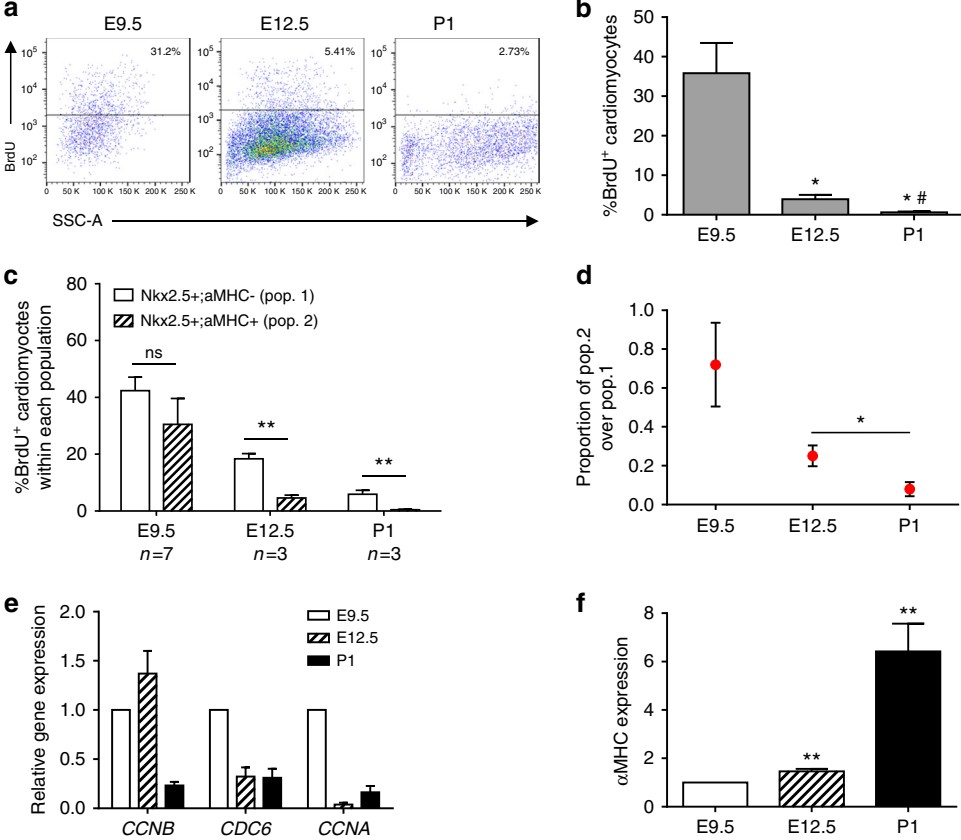

**Fig. 4** BrdU pulse-chase experiments substantiate decreasing proliferative capacity of CMs. **a** Representative flow cytometric analysis of BrdU incorporation. BrdU was given at E9.5, E12.5, or P1 αMHC-GFP mice 3 h prior to analysis. **b** Quantification of BrdU⁺ αMHC-GFP⁺ cells at E9.5, E12.5, and P1. (Student's t test), *E12.5 or P1 vs. E9.5, #P1 vs. E12.5, $p < 0.05$. **c** Quantification of percent BrdU incorporation in Nkx2.5⁺;αMHC⁻ and Nkx2.5⁺;αMHC⁺ cells at E9.5, E12.5, and P1. (Student's t test), **$p < 0.01$. **d** Proportion of BrdU⁺ cells within the cardiomyocyte (Nkx2.5⁺/αMHC⁺) compared to progenitor (Nkx2.5⁺/αMHC⁻) populations at E9.5, E12.5, and P1. (Student's t test), *$p < 0.05$. **e** qPCR analysis of αMHC-GFP+ cells reveals an age-dependent drop in expression of cell cycle genes (Ccna2, Ccnb1, Cdc6). **f** qPCR analysis of αMHC gene expression in αMHC-GFP+ cells from E9.5 to P1. All measurements shown are depicted as mean ± s.e.m

To further characterize the heterogeneity we observed in the heat map analysis of CMs from E9.5 and E12.5, we utilized a k-means clustering algorithm to classify subgroups within the t-SNE (Fig. 5d). This approach yielded four distinct clusters, which we used for subsequent transcriptomic analyses (Supplementary Figure 14a). Of particular interest are Clusters 1 and 3, which are enriched for E9.5 and E12.5 cells, respectively. To identify potential transcriptomic changes that occur between E9.5 and E12.5 that may contribute to the clonal size differences observed in our in vivo studies, we performed gene ontology analysis of differentially expressed genes between these clusters (Fig. 5e). Pathways involved in developmental growth, cell division, and migration were downregulated within Cluster 3 compared to Cluster 1. Conversely, there was upregulation in pathways involved in the regulation of heart contraction, cellular respiration, and muscle development within Cluster 3. Interestingly, we identified four genes (Thbs4, Kif26b, Col2a1, and Prtg) that were only expressed in E9.5 cells, and two others (Sall4 and Hmga2) whose expression was primarily concentrated in cells from this time point (44% and 53% of E9.5 cells, respectively) (Supplementary Figure 14b). From these, Thbs4, Sall4, and Hmga2 are genes associated with Gene Ontology pathways involved with developmental growth and cell division and migration. As expected, a comparison of genes enriched in Cluster 4 (containing primarily P1 cells), revealed an increase in pathways involved in cellular respiration, heart contraction, and metabolism when compared to Cluster 1 or 3. On the other hand, pathways involved in cellular proliferation and developmental growth were downregulated in cells within this cluster. These results suggest that by E12.5, CMs have already begun to shift to an initial state of maturation and have downregulated vital genes involved with cell division and migration, potentially contributing to the decrease in clone sizes observed in our analysis of CMs labeled at this developmental time point. Interestingly, we observed a P1 cell which segregated more closely with other E9.5 and E12.5 cells than their counterparts (residing within Cluster 3). Differential gene expression analysis revealed considerable differences in the gene profiles of this cell with the remaining P1 cluster (Supplementary Figure 14c). Analysis using the STRING (Search Tool for the Retrieval of Interacting Genes/Proteins) database comparing the top 50 genes more highly expressed in the P1 cluster compared to the outlier, identified two nodes associated with metabolism and cardiac structural proteins (Supplementary Figure 14d). This suggests that this particular P1 cell may be a rare CM present in the postnatal heart that retains some immature, more proliferative characteristics generally present only in earlier embryonic CMs. However, further studies examining a larger number of cells with similar characteristics are needed to substantiate this observation.

To examine how expression of cell cycle genes may relate with CM maturity, cells from E9.5 and E12.5 were selected based on low or high cycling activity, and concomitantly, their expression

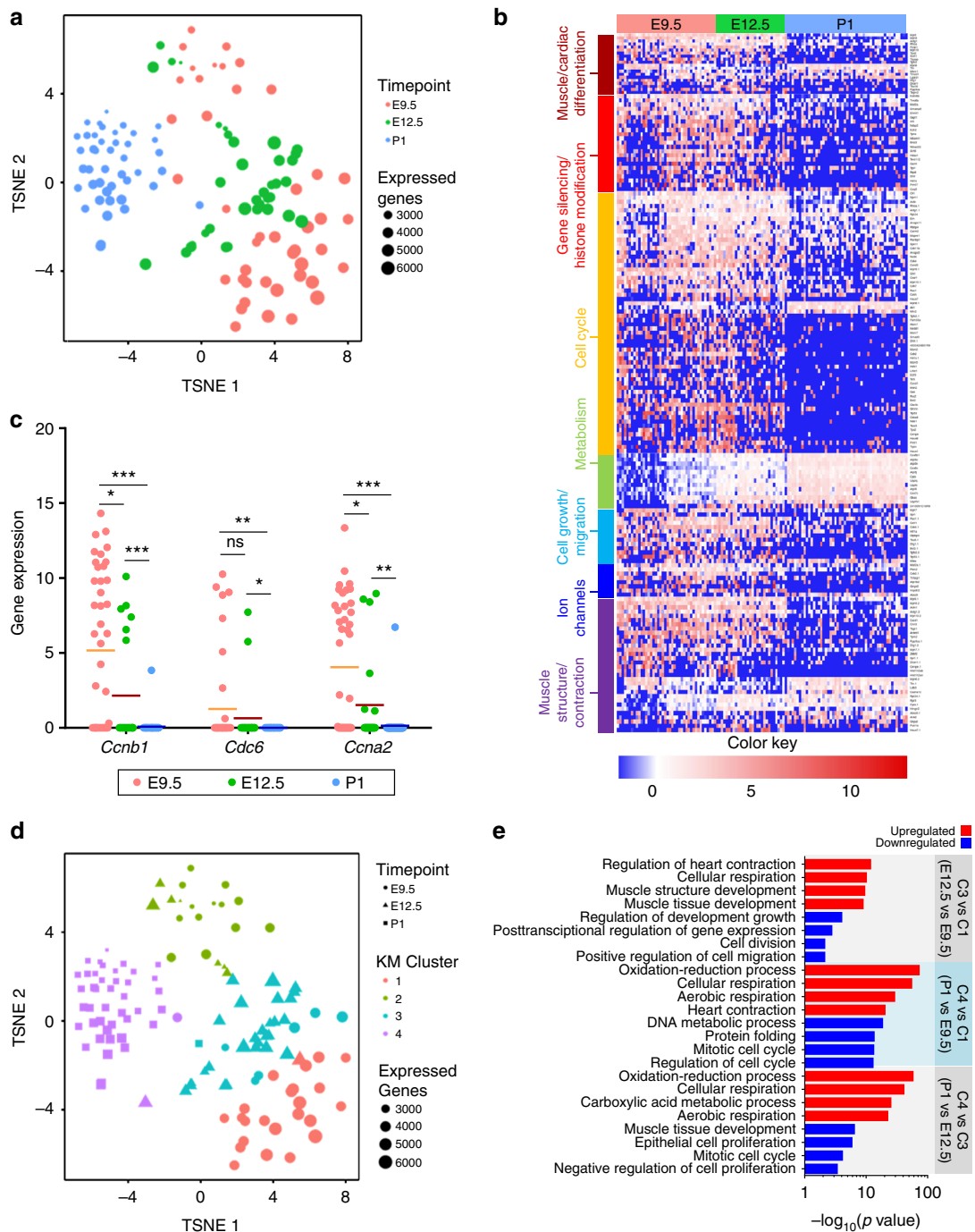

**Fig. 5** Single-cell gene expression analysis reveals heterogeneity of CMs within and between developmental time points. **a** t-SNE revealing distribution of single cells. **b** Heat map analysis of single cells from different developmental time points on genes relevant to cardiac development and maturation (E9.5: $n = 42$, E12.5: $n = 29$, P1: $n = 52$). The full list of genes presented in the maps is also provided in Supplementary Table 1. **c** Expression of cell cycle-associated genes (*Ccna2, Ccnb1, Cdc6*) from individual cells at each time point confirming a decline in cell cycle activity from E9.5 to P1 (Student's *t* test), *$p < 0.05$, **$p < 0.01$, ***$p < 0.001$. Mean depicted as solid line. **d** t-SNE with k-m-based clustering identifies four distinct clusters. **e** Gene ontology analysis of upregulated and downregulated genes between different clusters

of cardiac structural proteins and transcription factors was examined (Supplementary Figure 14e). CMs with high cell cycle activity according to gene expression profiles displayed a "progenitor-like" gene profile, in which there was low expression of structural proteins (*Myh6, Ttn, Myl4, Flnc*) indicating immaturity, and high expression of transcription factors involved in embryonic and cardiac development (*Nkx2-5, Gata6, Tbx18*). Likewise, CMs with low cell cycle activity had an expression signature indicative of

a more mature CM, with high levels of structural proteins and low levels of early cardiac transcription factors. These findings indicate a possible link between cell cycle activity and CM maturity that may be independent of embryonic age.

**Injury activates proliferation in neonatal but not adult CMs.** We next examined whether regeneration in neonatal injured

hearts is due to clonal expansion of αMHC-expressing CMs or non-αMHC (progenitor) cells. Newborn (P0) *αMHC*^CreER; *R26*^VT2/GK, *βactin*^CreER; *R26*^VT2/GK, and *Nkx2.5*^CreER; *R26*^VT2/GK mice received tamoxifen followed by left anterior descending artery (LAD) ligation or sham operation 24 h later (P1). At 21 days post-injury (dpi) the majority of the tissue was

regenerated and only a small fibrotic area remained (Fig. 6a–c; Supplementary Figure 13a). We observed frequent clones of CMs in the injury and border areas of *αMHC*^CreER; *R26*^VT2/GK hearts, indicating division of CMs in response to injury. CM clones were also detected in the remote zones, albeit at a lower number. Quantification of the number of clones consisting of two or more

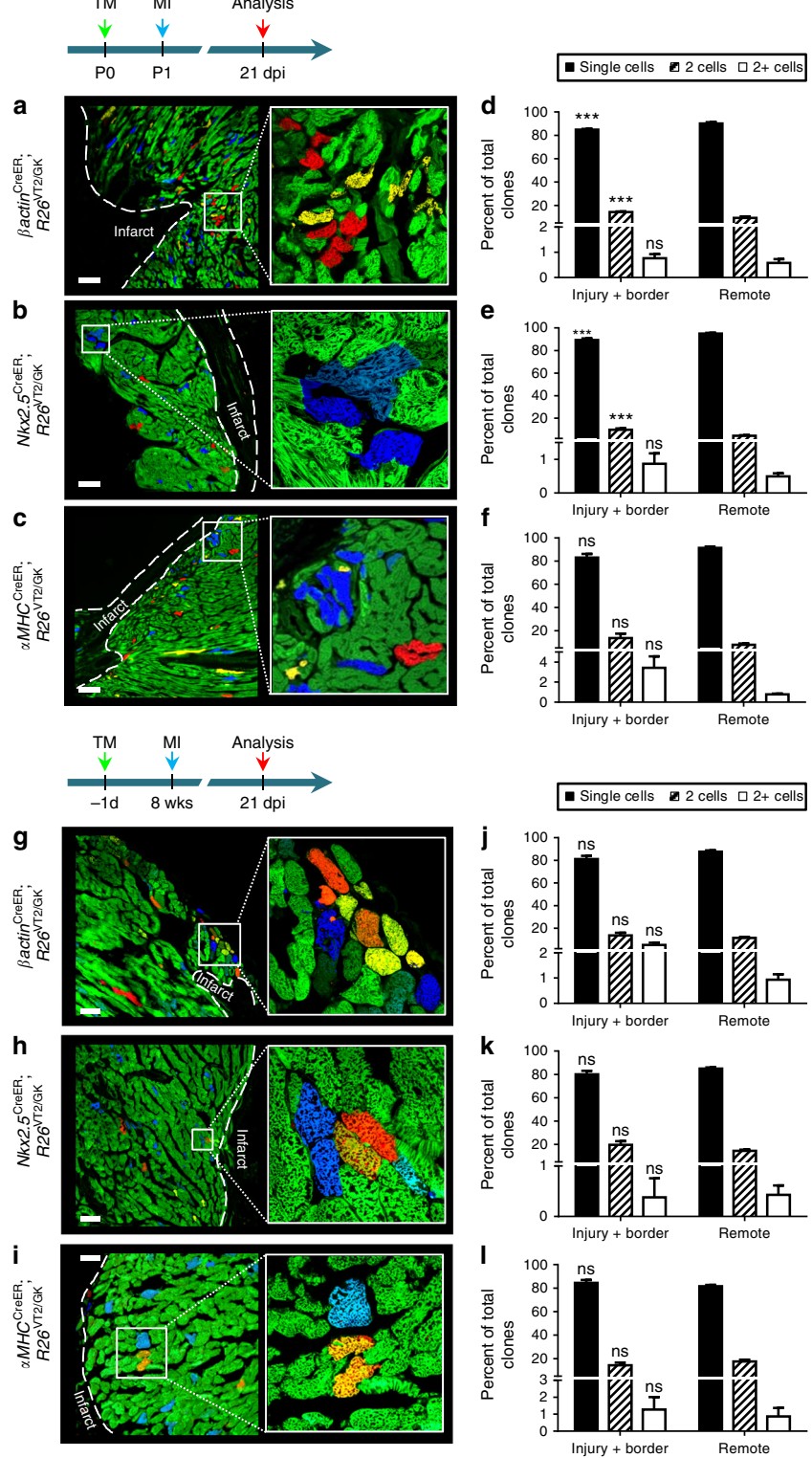

cells in these models did not reveal significant differences, indicating that regeneration is largely due to CM division (Fig. 6d–f). Our observations support previous studies showing CM proliferation following neonatal cardiac injury[20,21]. However, it should be noted that this model cannot exclude the possibility of de-differentiation or reversion of existing CMs into a more primitive state with subsequent proliferation.

We next sought to find clonal evidence for CM proliferation in adult hearts after injury. In contrast to neonatal mice, LAD ligation performed in adults resulted in scar formation with sparse single-labeled cells and very rare two-cell clones detected in the infarct border zone (Supplementary Figure 13b; Fig. 6g–i). Overall, we observed a similar pattern of mosaic labeling, with no evidence of clonal dominance, in $Nkx2.5^{CreER}$; $R26^{VT2/GK}$, $\beta actin^{CreER}$; $R26^{VT2/GK}$, or $\alpha MHC^{CreER}$; $R26^{VT2/GK}$ adult hearts after injury (Fig. 6j–l).

## Discussion

Stochastic labeling using the Rainbow model offers a precise, versatile method to retrospectively examine the proliferative behavior of CMs at a single-cell resolution. Utilizing this approach, we provide a direct and detailed investigation of mammalian embryonic and postnatal cardiovascular development. The existence of large clones observed during early development reflects the proliferative characteristic of progenitor cells that we also observed in early αMHC-expressing CMs. Our study points to the possibility that cardiac progenitors are able to maintain their proliferative potential for a longer span of time and contribute to a considerable portion of cardiac growth during embryonic development. Additionally, it is likely that their transition into a mature, less proliferative state occurs on a continuous, rather than discrete timescale with gradual expression of cardiac-specific markers concomitant with decreasing proliferative capacity.

Our BrdU and single-cell RNA sequencing experiments support the notion of heterogeneity within αMHC-expressing CMs, particularly within the early embryonic stages. The lower proliferative capacity at E12.5 compared to E9.5 may be due to combined effects of CM maturation: reduced response to developmental growth signals and cellular migration and a shift toward processes involved in heart contraction and cellular respiration. However, congruent with the idea of heterogeneity, we were able to identify CMs at E12.5 that still maintained high cell cycle activity and displayed a more "progenitor-like" gene profile.

Since our findings demonstrate the limited ability of postnatal CMs to undergo division, we sought to determine whether myocardial injury is sufficient to reactivate their proliferative capacity. It has been suggested that the mammalian heart retains a regenerative capacity shortly after birth that diminishes with age[20–22]. Our quantification of clone sizes within $\alpha MHC^{CreER}$, $R26^{VT2/GK}$, $\beta actin^{CreER}$, $R26^{VT2/GK}$, and $Nkx2.5^{CreER}$; $R26^{VT2/GK}$ mice after neonatal and adult injury suggests that CM proliferation can be activated following injury early in life and that this ability is significantly limited with aging.

Potential limitations of the current study include the possibility that administration of a minimal dose of tamoxifen may not induce recombination in a very rare subset of αMHC+ CMs with proliferative potential in later time points. Thus, it is possible for small clones of cardiomyocytes that were not captured in the system to make sizeable contribution to murine heart growth during this time. We also cannot exclude the possibility that a small progenitor cell population exists, capable of proliferation during adulthood (in normal aging or upon injury) that was not marked by our labeling strategy. However, this is unlikely given the consistency of our observations across a large number of samples as well as the enhanced CM proliferation observed following neonatal injury.

In conclusion, we have developed a robust in vivo mammalian model of clonal analysis to provide important and direct insights into the source of CMs and the characteristics of progenitor cells both during development and after injury. These findings are critical for ongoing efforts to develop regenerative therapies for cardiovascular diseases[23] in which an understanding of the mechanisms that regulate or reactivate the proliferative potential of these cells is a vital aspect.

## Methods

**Generation of Rainbow mice.** All animal studies were performed according to the guidelines of UCLA's animal care and use committee and the National Institutes of Health Guide for the Care and Use of Laboratory Animals. Both male and female mice were used for all animal experiments within this study. Rainbow transgenic mice (mixed background) were initially generated in I. Weissman's lab at Stanford University. Rainbow mice carry a cassette of 4 fluorescent proteins, inserted in the Rosa26 locus under the control of CAG promoter. Upon Cre-mediated recombination the default GFP expression is replaced by random expression of one of three other fluorescent proteins: mCherry, mOrange and mCerulean. $Mesp1^{Cre}$ ($Mesp1^{tm2(cre)Ysa}$), $Rosa26^{CreER}$ (B6.129-Gt (ROSA) 26Sortm1$^{(cre/ESR1)Tyj}$/J), $Nkx2.5^{Cre}$ (Nkx2-5$^{tm1(cre)Rjs}$), $\alpha MHC^{Cre}$ (Myh6-cre), $\alpha MHC^{CreER}$ (Myh11-cre/ESR1), $\beta actin^{CreER}$ (ACTB-cre/Esr1), and $Wt1^{CreER}$ (Wt1$^{tm2(cre/ERT2)Wtp}$/J) transgenic mice were obtained from The Jackson Laboratory. All animals were housed in sterile micro insulators and given water and rodent chow ad libitum.

**Generation of Nkx2.5$^{CreER}$ knock-in mice.** The targeting vector Nkx2.5IRESCre was a generous gift from Dr. Richard P. Harvey's laboratory[24]. It consisted of a gene cassette (IRES-Cre) carrying an internal ribosome entry site linked to the gene encoding a nuclear-localizing recombinase, which was inserted into the 3′ untranslated region (utr) of Nkx2.5. This vector was further modified by replacing the Cre sequence with the CreERT2 in-frame downstream of the IRES. The resulting construction vector included a hygromycin resistance gene cassette (pgk-HYGRO-pA) flanked by yeast flp recombinase target (frt) sites inserted downstream of IRES-CreERT2. The correct targeting occurred at a frequency of ~1 in 7 (22/152) hygromycin-resistant ES cell clones. Blastocyst injection of a single correctly targeted clone produced chimeric animals that passed the modified allele through the germline, generating the strain Nkx2.5IRESCreERT2HYGRO. To remove the pgk-HYGRO-pA cassette, founders were crossed with transgenic mice expressing the Flp recombinase gene (FLP1) in germ cells. Founders of this new strain (Nkx2-5IRESCreERT2) were backcrossed onto C57BL/6 mice. Validation and genotyping of mice were performed by Southern analysis and PCR. The resulting transgenic mice were fully viable, healthy, and fertile over several generations. In addition, this strategy resulted in preservation of the endogenous NKX2.5 loci, thereby avoiding haploinsufficiency, which may inadvertently influence developmental studies. Mice were genotyped by PCR using Cre-for (5′-TGCAGGTTTTGAGCCCTAAC-3′) and Cre-rev (5′-CGA-GAATGACTTCCCTGTCC-3′).

It should be noted that not all mouse models in this study were knock-ins, posing the potential limitations associated with use of ectopically inserted promoters.

**Fig. 6** Myocardial injury activates CM proliferation in neonatal but not adult mice. Neonatal mice underwent left anterior descending artery (LAD) ligation at P1 and clonal analysis performed 21 days post injury. Representative confocal images of **a** $\beta actin^{CreER}$; $R26^{VT2/GK}$, **b** $Nkx2.5^{CreER}$; $R26^{VT2/GK}$ and **c** $\alpha MHC^{CreER}$; $R26^{VT2/GK}$ heart (left ventricle) sections. Insets show close-up of boxed regions. **d-f** Quantification of clonal formation following neonatal injury. LAD ligation was performed in 8-week-old mice (**g**, **h**, **i**), followed by clonal analysis 3 weeks post-injury. Representative fluorescent microscope images of the infarct and border zone of **g** $\beta actin^{CreER}$; $R26^{VT2/GK}$, **h** $Nkx2.5^{CreER}$; $R26^{VT2/GK}$ and **i** $\alpha MHC^{CreER}$; $R26^{VT2/GK}$ hearts (left ventricle). **j-l** Quantification of clonal formation following adult injury. White dashed line marks infarct area. (Tukey's multiple comparison test), ***$p < 0.001$ to remote. All measurements shown are depicted as mean ± s.e.m

**Hydroxytamoxifen preparation and administration**. Tamoxifen (Sigma) was dissolved by sonication in corn oil to a stock concentration of 20 mg/ml. 4-hydroxytamoxifen (4OH-TM, Sigma) was first dissolved in absolute ethanol to a concentration of 100 mg/ml followed by dilution with corn oil to a final stock concentration of 10 mg/ml. Extensive sonication and vortexing were used. Stock solutions were used within 3 days of preparation. To achieve minimum recombination, one dose of tamoxifen was given through intraperitoneal injection. Pregnant females from $\alpha MHC^{CreER}$; $R26^{VT2/GK}$ and $\beta actin^{CreER}$; $R26^{VT2/GK}$ crossings received 4 mg and 350 µg of tamoxifen, respectively. Postnatally, $\alpha MHC^{CreER}$; $R26^{VT2/GK}$ mice received 10 µg and $\beta actin^{CreER}$; $R26^{VT2/GK}$ mice received 1 µg of tamoxifen. Pregnant females from $Wt1^{CreER}$; $R26^{VT2/GK}$ and $Rosa26^{CreER}$; $R26^{VT2/GK}$ crossings were treated with 4 mg of 4OH-TM and 350 µg of tamoxifen, respectively.

**Measurement of tamoxifen levels in serum**. Serum levels of 4OH-TM were measured after administration of tamoxifen to pregnant female mice. An aliquot of mouse plasma was mixed with acetonitrile in a 1:2 ratio, and then vortexed and centrifuged at 12,500 rpm for 8 min. The supernatant was removed and evaporated under a stream of argon. The resulting residue was reconstituted by vortex mixing with methanol. The solution was transferred to a tapered, limited volume autosampler vial which was then analyzed. The chromatographic separations were performed on a 10 cm × 300 µm Higgins Targa C18 3 µm column maintained at 50 °C. A 1 µl aliquot of sample was injected into the column and eluted with a 1–95% B gradient over 5 min where solution A was aqueous formic acid (0.1%) and solution B was acetonitrile. The eluate was analyzed and detected by tandem mass spectrometry (MS/MS) using selected reaction monitoring of the transition $m/z$ 388.24 ≥ 71.9 with a Waters Micromass LCT Premier MS that allows accurate detection to the 1 nM range.

**Tissue harvest and processing for histological analysis**. Hearts were harvested, perfused, and incubated in 4% (vol/vol) paraformaldehyde (4–6 h and 12–18 h at 4 °C for embryonic and postnatal hearts, respectively) followed by incubation in 30 % (wt/vol) sucrose in PBS at 4 °C for 12–18 h. The samples were removed from the sucrose solution and tissue blocks were prepared by embedding in Tissue Tek O.C.T. (Sakura Finetek). Blocks were kept frozen in −80°C. Frozen whole-heart blocks were sectioned at 7–10-µm-thick sections with a Leica CM1860 cryostat and mounted on Superfrost/Plus slides (Fisherbrand). Fluorescent images were acquired with Leica fluorescence inverted microscope DMI6000B equipped with an EL6000 light source and LED 590 (585/40), GFP ET (470/40), A4 ET (360/40), CFP ET (436/20) S-Gold (575/30) filter cubes. Confocal images were obtained with a Leica TCS-SP5 AOBS confocal multiphoton microscope. Lasers 458 nm, 488 nm (495 nm and 514 nm), and 594 nm were utilized to excite mCerulean, GFP, mOrange, and mCherry, respectively.

**Immunofluorescence**. Sections were washed three times with PBS followed by antigen retrieval in 0.25% Triton for 10 min. Samples were blocked for 1 h in 10 % goat serum followed by incubation with primary antibodies for 2 h at room temperature (RT). Antibodies against α-sarcomeric Actinin (1/400, Sigma, A7811), α-smooth muscle actin (1/100, Sigma, A2547), CD31 (1/100, Abcam, ab28364), DDR2 (1/100, R&D, MAB25381), WT1 (1/100, Abcam, ab89901), and PDGFRα (1/100, Santa Cruz, sc338) were used. Alexa fluor 647 secondary antibodies were used (1:100, Invitrogen) for 1 h at RT. Co-localization of proteins with Rainbow-labeled clones was obtained through confocal z-stack analysis. WGA (1/500, Invitrogen) staining was performed for 1 h at RT. For co-localization purposes, blue pseudocolor was assigned to all clones analyzed in this manuscript.

**In vitro clonal analysis**. Pregnant female mice were euthanized by isofluorane anesthetic overdose followed by cervical dislocation. The uterus containing E10.5 embryos was removed through an incision at the lower abdomen and was placed in sterile PBS supplemented with anti-biotic and anti-mycotic agents (PBS-AB/AM) at 37 °C. Embryos were dissected out of the uterus and their hearts harvested under a dissection microscope. Hearts were rinsed with PBS-AB/AM (37 °C) and digested in 0.2% (v/v) Collagenase type I solution for 1 h at 37°C. Following this, digested cells were collected by centrifugation (3 min, 3000 rpm) and resuspended in DMEM supplemented with 10% FBS. Cells were counted using a hemocytometer and plated onto 12-well culture dishes at a density of 500,000 cells per dish. Cells were maintained at 37 °C in 6% $CO_2$ for 120 h. Images of single cells labeled with a Rainbow color were acquired with a Leica fluorescence microscope every 8 h. The feature "mark and find" of the Leica AF6000 software was used to ensure precise repositioning of defined positions within the culture dish.

**In vivo clonal analysis**. The offspring of crossings between $R26^{VT2/GK}$ and $Mesp1^{Cre}$, $Nkx2.5^{Cre}$, or $\alpha MHC^{Cre}$ were analyzed at P1, P14, and P21 ($n = 4$ for each). To determine clonal expansion in $\alpha MHC^{CreER}$; $R26^{VT2/GK}$ and $\beta actin^{CreER}$; $R26^{VT2/GK}$ mice, tamoxifen was administered at E12.5 and E9.5, respectively and animals were analyzed at P2, P7, P15, and P30. In a second set of experiments, tamoxifen was administered at E12.5, P2, P7, and P15 and analysis was performed at P30. For experiments with $Wt1^{CreER}$; $R26^{VT2/GK}$ mice 4OH-TM was administered at E9.5 and analysis was performed at P0. Pregnant females from

$Rosa26^{CreER}$; $R26^{VT2/GK}$ crossings received tamoxifen at E7.5 and were analyzed at P2. For each mouse strain and timepoint $n = 6$.

Images were processed with the Leica AF6000 software. 2D images of every ten sections of the whole tissue were obtained for each timepoint and experimental condition. Sections were stained with WGA to identify cell boundaries (Supplementary Figure 13). To specifically distinguish CM clones, sections were stained with α-sarcomeric actinin and co-localization of actinin with each clone was verified. Quantification of clone number and size was performed by manual counting of the number of single-color clusters and the cells within each cluster. Sample identity was hidden from the operator in order to perform quantifications in a blinded fashion.

**Clearing of P2 mouse hearts using CLARITY for 3D quantification of clone volumes**. P2 mouse hearts were rinsed in PBS and then placed in 4% paraformaldehyde (Electron Microscopy Sciences) at RT for 4 h. The tissues were then rinsed in PBS and transferred to a 4% acrylamide solution (Bio-Rad) along with 0.5% w/v of the photoinitiator 2,2'-Azobis[2-(2-imidazolin-2-yl)propane]dihydrochloride (VA-044, Wako Chemicals USA) and incubated overnight at 4 °C. To initiate polymerization, tissues were incubated in the same solution at 37 °C for 3 h. After polymerization, the tissues were rinsed with PBS and then placed into a solution of 8% w/v sodium dodecyl sulfate (Sigma Aldrich) and 1.25% w/v boric acid (Fisher), pH 8.5 at 37 °C until cleared. Tissues were transferred to PBS for one day to remove residual SDS and stored in refractive index matching solution (RIMS) until imaging. The RIMS formulation is as follows: to make 30 ml RIMS, dissolve 40 grams of Histodenz (Sigma) in 0.02 M Phosphate buffer (Sigma) with 0.05% w/v sodium azide (Sigma) and syringe filter through a 0.2 mm filter. The tissues can be stored at RT in RIMS until ready for imaging.

**3D imaging and reconstruction**. Light sheet fluorescent imaging was carried out on a system previously developed[25]. A diode-pumped solid-state laser containing four wavelengths of 405 nm, 473 nm, 532 nm, and 589 nm (LMM-GB1, Laserglow Technologies, Toronto, ON, Canada) was used as the illumination source. The laser beam was first expanded from its initial diameter of 2 mm to 10 mm using a 5 × achromatic beam expander (GBE05-A, Thorlabs Inc, New Jersey, USA). Then the expanded beam was focused by a plano-convex cylindrical lens (=50 mm, LJ1695RM-A, Thorlabs Inc, New Jersey, USA) and was then reshaped by a group of achromatic doublets (AC254-060-A, AC254-100-A, Thorlabs, New Jersey, USA). After passing through an $f = 150$ mm lens, a laser-sheet 40 mm wide and 17 µm thick (Full width at half maximum value) was generated to optically section the entire heart sample. The detection path including an objective lens (2 × /0.06, Nikon), a tube lens (ITL 200, Nikon) and switchable optical filters (Semrock, New York, USA) was placed orthogonal to the illumination path for collecting the fluorescence signals. A scientific CMOS (ORCA-Flash4.0 V2, Hamamatsu, Japan) was mounted at the terminal to record the digitalized images with a high frame rate. Samples were mounted into a borosilicate glass tubing and placed on a motorized translational stage. The sample and its holderwere immersed in a chamber filled with 99.5% glycerol. Illumination and detection were computer-controlled. Horizontal stripe shadow artifacts were removed from light sheet images in the Fast Fourier Transform image of each plane by masking the frequency of the stripes near the $Y$ axis using FIJI ImageJ software[26]. The images of serial sections were then assembled into a z-stack and processed. The contrast was adjusted and adjacent sections were automatically aligned using the StackReg plugin in Rigid Body mode[27]. A 3D volume of the aligned z-stack was rendered using Imaris x64 *ver. 7.7.1 (Bitplane AG, Zurich). Single clones were selected using the Isosurface function of Imaris.

**Neuregulin treatment**. Recombination was induced at P15. Starting at P20 $\alpha MHC^{CreER}$; $R26^{VT2/GK}$ mice received daily injections of Neuregulin (2.5 µg/mouse, NRG1, R&D) or control BSA (0.1%) for 9 days. Analysis was performed at P29 (control $n = 3$, NRG1 $n = 4$). Animals received BSA or NRG1 in a non-randomized manner.

**BrdU pulse and chase experiments**. Dames carrying $\alpha MHC$-GFP embryos at E9.5 or E12.5 and P1 neonates were administered 0.1 mg BrdU/g body weight of BrdU (BD Biosciences, 550891). Three hours post injection, hearts were collected and digested in collagenase type II (2600 U/ml) for 30 min at 37 °C and washed with 1 ml FACS Buffer. Cells were centrifuged and supernatant removed. BrdU staining was performed according to manufacturer's protocol (BD Pharmingen, 552598) and cells were FACS analyzed using BD LSRFortessa™ Cell Analyzer Systems.

**Single-cell RNA sequencing using Fluidigm C1 platform**. $\alpha MHC$-GFP mouse hearts at E9.5, E12.5, and P1 were harvested and digested for FACS sorting. Hearts were digested in either 0.2% (v/v) Collagenase type I solution for 30 min at 37 °C (E9.5 and E12.5) or with Liberase Blendzyme TH and TM (Roche) in Medium 199 with DNAase I and polaxamer for 1 h at 37 °C (P1). P1 samples were passed through a 70 µm cell strainer (BD Falcon) before centrifugation at 450 × $g$ for 5 min, supernatant aspirated, and pellet resuspended in FACS buffer. Single-cell RNA sequencing using the C1 Fluidigm platform was performed by the Genomics

Core at Cedars-Sinai Medical Center. Cells were sorted directly into C1 Suspension Reagent to obtain a range of 200–260 cells/µl and 6–8 µl of cell suspension were loaded according to the manufacturer's protocol on a primed C1 Single Cells AutoPrep Medium IFC microfluidic chip. The chip was run on a Fluidigm C1 instrument and images of all 96 capture sites were taken on a Leica DMi8 Fluorescent Microscope at ×100 and ×200 magnification. Lysis, reverse transcription, and PCR was then performed using the Fluidigm "mRNA Seq: RT + Amp (1772 × /1773 × )" script and commercially available kits from Life Technologies (Life Technologies Superscript II reverse transcriptase) and Clonetech (dSMART-Seq V4 Kit) according to manufacturer's instructions. Amplified cDNA was harvested in a total of 13 µl of C1 Harvesting Reagent and quantified on an Agilent Bioanalyzer. Sequencing was performed on the NextSeq on a $1 \times 75$ high output flow cell following the NextSeq denature and dilute guide (Protocol A: Standard Normalization Method).

**Single-cell RNASeq analysis**. RNASeq data was mapped with OLego version 1.1.5[28] and normalized by using TPM (Transcripts per millions) analysis as shown in Supplementary Figure 14f. Total number of reads mapped to a known transcript annotation was estimated using featurecCounts version v1.5.0-p2[29]. Expression levels for each transcript were determined by normalizing the counts returned by featureCounts using custom Perl scripts. Normalized expression levels for each transcript were determined by transforming the raw expression counts to TPM following log2 scaling. Total number of reads mapped to the genome showed a bimodal distribution. This bimodal distribution separated high quality data and low-quality data. We performed StepMiner[30] analysis to determine a threshold to identify cells with high quality data. All the cells with low-quality data were removed from subsequent analysis. Clustering was performed using standard hierarchical agglomerative approach using Euclidian distance as a metric for complete linkage (Fig. 5b; Supplementary Figure 14f, R package "hclust").

t-SNE and differential gene expression analyses were performed as follows. Raw count matrices were generated using featureCounts and normalized by the number of mapped reads in log space (ln(mapped_reads/10,000 + 1)). We filtered to exclude low-quality cells with <2000 expressed genes (64 cells) and lowly expressed genes that are expressed in less than three cells (9714 genes) from all downstream analyses. This results in a normalized expression matrix of 13,623 genes among 122 cells (Supplementary Fig. 14a). Cells were projected onto a 2D embedding using t-Distributed Stochastic Neighbor Embedding (t-SNE, perplexity set to 27) with cell loadings associated to 30 principal components utilizing all expressed genes as input (Fig. 5a, R packages 'irlba' and 'Rtsne'). Cell cluster assignments were computed using K-means clustering with $k = 4$ (Fig. 5d). The number of K-means clusters was determined by computing the sum of squared error (SSE) for $k = 1…15$ and observing an "elbow" where the estimated number of clusters is observed. We implemented a negative binomial generalized linear model to identify differentially expressed genes enriched in each cluster. Genes satisfying an abs(log (average expression difference)) >0.5 and $p < 0.01$ were considered statistically significant. Gene Ontology enrichments among cluster enriched, differential genes were computed using Metascape (http://www.metascape.org) (Fig. 5e). RStudio (https://www.rstudio.com/) was used to run custom R scripts to perform the analyses described above. Generally, ggplot2 and pheatmap packages were used to generate data graphs.

**Myocardial infarction model**. $\beta actin^{CreER}$; $R26^{VT2/GK}$, $\alpha MHC^{CreER}$; $R26^{VT2/GK}$, and $Nkx2.5^{CreER}$; $R26^{VT2/GK}$ mice received tamoxifen 1 day prior to surgery. All surgical procedures were performed by a single, experienced surgeon, blinded to the identity of the mice. Myocardial infarction was generated via permanent ligation of the LAD. Neonatal mice (P1) ($n = 3$) were anesthetized in an isofluorane chamber (3% isofluorane) and placed on an ice bed for the entire procedure. An incision was performed at the fourth intercostal space and LAD was permanently ligated with an 8-0 suture. The chest wall was sutured with a 6-0 prolene suture and the skin wound was closed using a surgical skin adhesive. Following the procedure mice were placed under a heat lamp until recovery. Adult mice (8 weeks old) ($n = 5$) were anesthetized by intraperitoneal injection of ketamine (100 mg/kg) and xylazine (10 mg/kg). Animals were ventilated with oxygen-enriched room air during the entire procedure. Left thoracotomy was performed through an incision between the fourth and fifth intercostal muscles followed by removal of the pericardium. An 8-0 silk suture was used to permanently ligate the LAD. Post-operative discomfort was treated with buprenorphine (0.03–0.06 mg/kg). Sham-operated mice were submitted to the same procedure lacking the LAD ligation ($n = 3$ for P1 and 8 weeks old). Animals were submitted to sham operation or LAD ligation in a non-randomized manner.

**Statistical analysis**. Student unpaired $t$ test and one-way ANOVA were used for statistical analysis. All data are presented as mean ± SEM. Two-sample Kolmogorov–Smirnov distribution test was used to determine significance for 2D clonal size and 3D clonal volume analyses. A probability value $p \leq 0.05$ was considered statistically significant. All analyses were performed with GraphPad Prism 5.04.

**Data availability**. The authors declare that all data supporting the findings of this study are available within the article and its supplementary information files or from the corresponding author upon reasonable request. Sequencing and code sources that support the findings of this study have been deposited in GitHub at the following link: https://github.com/ShanSabri/Rainbow-analysis. Raw data files have been deposited in the GEO database under accession code GSE10842.

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

## Acknowledgments

We would like to thank Dr. Matt Schibler for assistance with confocal microscopy, completed at the UCLA CNSI Advanced light microscopy/Spectroscopy Shared Resource Facility, supported by NIH-NCRR shared resources grant (CJX1-443835-WS-29646) and NSF Major Research Instrumentation grant (CHE-0722519). We also thank Dr. Esteban Fernandez for his help with 3D reconstruction and video processing, Dr. Cheng-Han Chen for help with tamoxifen serum levels measurements and all Ardehali lab members for discussions and suggestions. This work was supported in part by grants from the California Institute for Regenerative Medicine (CIRM) (RN3-06378) (R.A.), the National Institutes of Health (NIH) DP2 HL127728 (R.A.), the American Heart Association (AHA-BGA 12BGIA8960008) (R.A.) and Eli & Edythe Broad Center of Regenerative Medicine and Stem Cell Research at UCLA Research Award (R.A. and H.M.), Scholars in Translational Medicine Program (R.A.), and UCLA BSCRC-Rose Hills Foundation Training Program (S.S.), the American Heart Association (14GRNT20480340) (H.M.), NIH R00-CA151673 and Department of Defense grant (W81XWH-10-1-0500) (D.S.), CIRM training grant (TG2-01169) (K.I.S.), and the Sarnoff Cardiovascular Research Foundation (N.B.N.).

## Author contributions

R.A., K-I.S., N.B.N., and P.K. conceived and designed the project, and wrote the manuscript. K-I.S., N.B.N., and P.K. performed experiments, analyzed data, and interpreted results with R.A., H.M., and M.L.I.A. P.Z. performed the animal surgeries, S.R. assisted with embryo extractions and single-cell RNAseq analysis. S.P. assisted with the proliferation assays. K.S. and R.P.K. helped with CLARITY. T.K.H. and Y.D. assisted with image acquisition and analysis. D.S., S.S., K.P., and J.E. helped with single-cell gene expression analysis. J.L.E. helped with image acquisition and clone counting.

## Additional information

**Competing interests:** The authors declare no competing financial interests.

