## [Peer Review File · Nature Communications]

Reviewers' Comments:

Reviewer #1:

Remarks to the Author:

In this beautifully done study Seriti et al examined the generation of cardiomyocytes during mouse embryonic and fetal development using cell-tracing approaches. The authors employed various Cre recombinase drivers together with the Rainbow system to analyze the contribution of cardiac progenitor and myosin heavy chain expressing cardiomyocytes to the myocardium at different developmental stages. Not very surprisingly, the author found that cardiac progenitor cells generated larger clones compared to cardiomyocytes at the same developmental stage and that cardiomyocytes at late (E12.5) developmental stages formed smaller clones than at early developmental stages (E9.5). The decreasing proliferative capacity of cardiomyocytes over time was further substantiated by BrdU pulse/chase experiments. Furthermore, the authors attempted to find a molecular correlate for the reduced proliferative potential of cardiomyocytes by performing single cell RNA sequencing. Again not very surprisingly, the authors disclosed that genes encoding for cell cycle etc. were downregulated in P1 compared to E9.5 and E12.5 cardiomyocytes whereas structural and metabolic genes were upregulated. Although E9.5 and E12.5 cardiomyocytes could be distinguished in a PCA, cells in this group show a high heterogeneity, which made it difficult to understand the differential proliferative capacity.

The study is based on state-of-the-art techniques and provides a convincing framework of the contribution of cardiac progenitor and cardiomyocytes to cardiac growth during mouse development. Several of the findings are not very surprising but rather predictable if not to say trivial. Cardiac progenitor cells are known to be highly proliferative in contrast to cardiomyocytes, which grow more slowly. Hence, one would expect larger clones from progenitor cells compared to cardiomyocytes. Nevertheless, in my view it is important to demonstrate such differences *in vivo*, which the authors did in a very convincing manner. Similarly, it is not surprising that cardiomyocytes labeled at an earlier time point (E9.5) generate larger clones than cardiomyocytes labeled at a later time point and that "older" cardiomyocytes are more mature in terms of gene expression transcribing fewer cell cycle genes and at lower levels. Unfortunately, the authors did not analyze the relative contribution of progenitor and cardiomyocyte proliferation to absolute cardiac, which is a real shortcoming. In addition, there are a few technical concerns.

Specific comments

The authors nicely demonstrated that cardiomyocytes at E12.5 form smaller clones compared to E9.5 (and much smaller clones compared to cardiac progenitor cells at E12.5). The overall contribution, however, of progenitor and cardiomyocyte cell proliferation to cardiac growth at different developmental stages remains unclear. A large number of cardiomyocytes, which divide only once or twice might generate a huge part of the myocardium whereas the contribution of a low number of large clones derived from cardiac progenitor cells might be rather low. With other words: is the absolute cardiac growth at E12.5 really still driven by expansion of cardiac progenitor cells or by dividing cardiomyocytes, which might not form large clones but compensate for that "shortcoming" by their sheer number? I am not sure whether the authors can address this question with their technical approach, which is based on stochastic, incomplete labeling of different cell populations making quantification of overall contributions difficult.

The single cell sequencing experiments of myosin heavy chain positive cells at E9.5, E12.5 and P1 are interesting. However, only a rather small number of individual cells (123 cells) was analyzed (i.e. only ca. 40 cells per time point). Moreover, the transcriptional analysis did not identify clear distinguishing features between E9.5 and E12.5 cardiomyocytes although the latter gave rise to smaller clones. On the other hand E9.5 and E12.5 cardiomyocytes showed a clearly distinct yet overlapping pattern in the PCA. I wonder whether analysis of a larger cohort or cells might identify a group of genes that allow E9.5 cardiomyocytes to form larger clones. The total number of analyzed cells (123 cells) is very small and should be enlarged, if possible. The authors should also

analyze sequencing data by t-SNE instead of PCA, which notoriously fails to reveal non-linear relationships.

When analyzing regeneration of neonatal hearts the authors mention that they also used the Nkx2.5 driver but did not show the results. Did the authors observe any Nkx2.5-CreER driven clones during neonatal heart regeneration? Did the clones occur at the same or lower frequency than alphaMyHC-CreEr driven clones? Was the clone size similar to alphaMyHC-CreEr driven clones? What happens if the Nkx2.5 driver is activated DURING regeneration?

The author state that the 3D analysis corroborated the 2D-analysis but described an about 16-fold drop of the size of alphaMyHC-CreEr driven clones between E9.5 and E12.5, which does not seem to correspond to the numbers obtained by the 2D-analysis. This should be clarified.

To trace cardiogenic cell clones the authors used different drivers. Some of the drivers were transgenic insertions while other were knock-ins. Since the authors performed a comparative analysis, it would have been more consistent to use knock-in alleles in all cases. I do not assume that the use of transgenic insertions creates major problems but the authors should briefly mention that transgenic promoter insertions were used in some cases.

Reviewer #2:

Remarks to the Author:

In their manuscript, Sereti et al use a series of Cre drivers and a multicolor lineage tracing system to systematically assess the proliferative potential of cardiomyocytes through development, neonatal heart regeneration, and following adult MI. The images are beautiful and the amount of work to generate this data is impressive. However, the rigor of the conclusions would be strengthened by consultation with a biostatistician with experience in cellular modeling.

Sereti et al's major conclusions are cardiac progenitors (defined by the absence of Myh6 labeling) contribute to the majority of cardiomyocytes during development and that the capacity for cellular proliferation is inversely related to the increased differentiation of cardiomyocytes. My guess is that most developmental biologists believe cellular proliferation and differentiation are inversely related. For the heart, I don't think anyone has really proven or quantified this concept, and Sereti et al go there.

Major Comments:

1. The results from Figure 3 are used to demonstrate the concept that more differentiated cardiomyocytes have a decreased capacity for proliferation than less differentiated progenitors. However, it also looks like fewer clusters were observed in aMHC:CreER analyses compared to the other strains. The aMHC:CreER is notoriously less efficient than the other Cre strains and the results of Sereti et al could also be the result of sampling fewer cardiomyocytes. The authors could address this concern by providing data on the total number of cardiomyocytes assessed with each Cre strain and formal statistical evaluation for a difference in the distributions of Fig 3 a, c, e, and g.

2. In their abstract, the authors claim, " Our data suggest that clonal dominance of differentiating progenitors mediates cardiac development, while a population of "immature" cardiomyocytes with a distinct molecular signature maintains homeostasis during late embryonic development and after birth." I don't think they really assessed homeostasis. I also don't think the authors demonstrated the presence of a population of "immature" cardiomyocytes in the postnatal heart. There was one outlier "immature" cell at P1 from the single cell sequencing experiments. Not sure a whole lot can be stated based on one outlier cell. The authors should remove references to cardiac homeostasis in the text.

3. I'd like to see the post MI adult data quantified. Are the rare clusters that are observed above the numbers of clusters that would be seen by just labeling an uninjured heart with the same conditions? Lots of investigators have been pointing to multicolor lineage tracing as an approach for identifying postnatal cardiomyocyte proliferation, but this looks to be a lot harder than theorized.

Minor Comments:

1. Figure 1 is used to develop the idea that progenitors are more proliferative than differentiating cells. However, with the labeling strategy, the patterns of clonal dominance seem to also reflect when the Cre is active. I would expect a Cre strain active during early development to result in larger clones. Since the authors used timed Cre induction to refine their argument, my comment is quite minor but probably should be addressed in the text.

2. Can the authors confirm that the clusters labeled by β -actin::CreER experiments in Figure 3 are all clusters of cardiomyocytes and not other cell types (fibroblasts, etc)? The authors should probably address this concern in their methods section.

3. Figures S3A and S3B are not convincing for overlap of the blue clone with DDR2 or CD31. Is there a better image?

4. The references in the text to Figures S8B and S8C seem to be one panel off.

5. In Figure 5a, are the numbers of observed clones significantly different at the regenerate versus a more remote area?

We appreciate all of the helpful suggestions we have received from the reviewers. We have addressed all concerns raised by the reviewers, which has greatly strengthened the revised manuscript. To summarize, we have made the following major changes:

1. We have consulted with a bio-informatician to re-analyze our single cell data. Using t-SNE, k_m based clustering, and gene ontology analysis, we now provide specific insights into the transcriptional changes that occurs between E9.5 and E12.5 cardiomyocytes, which may contribute to the clonal size differences we observed in our *in vivo* clonal analyses studies.
2. We have performed additional experiments and clonal size quantification of Rainbow mice crossed with different promoters to understand the proliferative capacity of cardiomyocytes under normal development and after injury.
3. We have taken a different statistical approach to determine significance of our clonal analysis studies.
4. We have improved the quality of select images to show co-localization of cell-surface markers of interest within a clone.

Below is our specific response to each of the reviewer's questions/concerns.

Reviewer #1 (Remarks to the Author):

1. Is the absolute cardiac growth at E12.5 really still driven by expansion of cardiac progenitor cells or by dividing cardiomyocytes, which might not form large clones but compensate for that "shortcoming" by their shear number? I am not sure whether the authors can address this question with their technical approach, which is based on stochastic, incomplete labeling of different cell populations making quantification of overall contributions difficult.

As indicated by the reviewer, this is an interesting but difficult question to examine thoroughly with our system. Although our 2D and 3D analysis of clone volume suggests that cardiac progenitors (marked by Nkx2.5) may have a higher proliferative rate than cardiomyocytes (marked by α MHC), we cannot claim that cardiac expansion is only regulated by cardiac progenitors at this time point. Nonetheless, we do observe cycling cardiomyocytes, labeled at E12.5, which undoubtedly contribute to cardiac growth during embryonic and perhaps neonatal period. As pointed out by the review, this effect may be further augmented by the higher number of cardiomyocytes at this developmental stage. However, due to the nature of tamoxifen titration to induce minimal recombination necessary for clonal analyses, we are unable to rule out the possibility that absolute cardiac growth can be driven by dividing cardiomyocytes at that time point, based on shear number. Our data suggest that proliferating cardiac progenitors along with slowly dividing cardiomyocytes jointly contribute to cardiac growth at E12.5. While this is an interesting question to address, we feel that this is out of the scope of our paper, which is focused primarily on examining the proliferative capacity of individual cardiomyocytes at different developmental stages.

2. Only a rather small number of individual cells (123 cells) was analyzed (i.e. only ca. 40 cells per time point). I wonder whether analysis of a larger cohort or cells might identify a group of genes that allow E9.5 cardiomyocytes to form larger clones. The total number of analyzed cells (123 cells) is very small and should be enlarged, if possible.

Despite the low number of cells which passed quality screening, we believe that it is sufficient for the fundamental analysis of the similarities/differences in these populations. Specifically, we were able to identify >862 differentially expressed genes between E9.5 and E12.5, >2260 between P1 and E9.5, and >1870 between P1 and E12.5. These numbers indicate that despite the low sample size, we are able to extract differences between these populations, acceptable for the level of analysis we wanted to pursue. While recent technologies have allowed researchers to perform single cell RNA sequencing for cell numbers up to the thousands, the approach we utilized at the time of our initial experiments (Fluidigm system) had limited capture efficiency and capacity. In addition, the heart at embryonic stages E9.5 and E12.5 contain low numbers of α -MHC⁺ cardiomyocytes and thus, we were able to achieve sufficient cell numbers through timed-mating and pooling of up to 8 hearts at each time point. In order to repeat the experiment and increase the number of single cells for RNA sequencing, we would need to use Fluidigm platform at another facility since the core facility at our institution is no longer available. In addition to the extended time required for the repeat single cell analysis, the cost would be in excess of \$10,000. As suggested by the reviewer and the editor, we performed in-depth analysis of our data to address the concerns raised by the reviewers. Through collaboration with a bioinformatics group at UCLA, we are now able to provide additional transcriptional insights from our original manuscript regarding the ability of cardiomyocytes at E9.5 to form larger clones. Please see our responses to comments 3 and 4 below.

3. Moreover, the transcriptional analysis did not identify clear distinguishing features between E9.5 and E12.5 cardiomyocytes although the latter gave rise to smaller clones. On the other hand E9.5 and E12.5 cardiomyocytes showed a clearly distinct yet overlapping pattern in the PCA.

As indicated by the reviewer, the distribution of E9.5 and E12.5 cardiomyocytes within the PCA is rather interesting. We observed clearly distinct subpopulations of these two groups, in addition to a region of overlap which we had denoted as Overlap in the original manuscript. We believe that analysis of these different subpopulations provide insights into the transcriptional changes regulating cardiomyocyte clonal growth. Per the helpful suggestion to utilize t-SNE (from comment 4 below), we were able to identify four clusters through a statistically accepted algorithm, providing us with similar subpopulations as we had originally identified through PCA analysis. However, this approach allowed us to compare more relevant populations and parse out new information regarding the pathways differentially enriched between the E9.5 and E12.5 cardiomyocytes. We have made changes to the sections indicated below.

Original Text: pg 9, line 5

Overall, we analyzed a total of 123 cells that passed quality screening. A combined principal component analysis (PCA) identified three clusters of cells that corresponded to their developmental time point (Fig 4C). A combined principal component analysis (PCA) identified three clusters of cells that corresponded to their developmental time point (Fig 4C). Heat map analysis of these clusters on genes relevant to cardiac development and maturation showed that P1 cardiomyocytes displayed a more mature, less proliferative transcriptional profile that is distinguishable from E9.5 and E12.5 clusters (Fig 4D).

Amended Text: pg 9, line 3

Overall, we analyzed a total of 122 cells that passed quality screening. Unsupervised dimensionality reduction by t-SNE identified clusters of single cells that appeared to correspond to their developmental time point (Fig 4a). Heat map analysis of these cells on genes relevant to cardiac development and maturation showed that P1 cardiomyocytes displayed a more mature, less proliferative transcriptional profile that is distinguishable from E9.5 and E12.5 clusters (Fig 4D).

Original Text: pg 9, line 25

To further characterize the heterogeneity we observed in the heat map analysis of cardiomyocytes from E9.5 and E12.5, we identified subgroups within each PCA cluster that appeared to have distinct transcriptome profiles.

Amended Text: pg 9, line 21

To further characterize the heterogeneity we observed in the heat map analysis of cardiomyocytes from E9.5 and E12.5, we utilized a k-means clustering algorithm to classify subgroups within the t-SNE (Fig 4d). This approach yielded four distinct clusters, which we used for subsequent transcriptomic analyses (Extended Fig 14c). Of particular interest are Clusters 1 and 3, which are enriched for E9.5 and E12.5 cells, respectively. To identify potential transcriptomic changes that occurs between E9.5 and E12.5 that may contribute to the clonal size differences observed in our *in vivo* studies, we performed gene ontology analysis of differentially expressed genes between these clusters (Fig. 4e). Pathways involved in developmental growth, cell division, and migration were downregulated within Cluster 3 compared to Cluster 1. Conversely, there was upregulation in pathways involved in the regulation of heart contraction, cellular respiration, and muscle development within Cluster 3. Interestingly, we identified 4 genes (*Thbs4*, *Kif26b*, *Col2a1*, and *Prtg*) that were only expressed in E9.5 cells, and 2 others (*Sall4* and *Hmga2*) whose expression was primarily concentrated in cells from this time point (44% and 53% of E9.5 cells, respectively) (Extended Fig 14e). From these, *Thbs4*, *Sall4*, and *Hmga2* are genes associated with Gene Ontology pathways involved with developmental growth and cell division and migration. As expected, a comparison of genes enriched in Cluster 4 (containing primarily P1 cells), revealed an increase in pathways involved in cellular respiration, heart contraction, and metabolism when compared to Cluster 1 or 3. On the other hand, pathways involved in cellular proliferation and developmental growth were downregulated in cells within this cluster. These results suggest that by E12.5, cardiomyocytes have already begun to shift to an initial state of maturation and have downregulated vital genes involved with cell division and migration, potentially contributing to the decrease in clone sizes observed in our analysis of cardiomyocytes labeled at this developmental time point.

Original Text: pg 10, line 8

Interestingly, the P1 outlier displayed an opposing transcriptional profile for some cell cycle genes and transcription factors compared to their P1 counterparts, and instead shared some resemblance to that of E9.5C2 cluster.

Amended Text: pg 10, line 13

Interestingly, we observed a P1 cell which segregated more closely with other E9.5 and E12.5 cells than their counterparts (residing within Cluster 3). Differential gene expression analysis revealed considerable differences in their gene profile (Supplemental Fig 14g).

Original Text: pg 10, line 13

This suggests that the P1 outlier may be a rare cardiomyocyte present in the postnatal heart that retains some immature, more proliferative characteristics generally present only in earlier embryonic cardiomyocytes.

Amended Text: pg 10, line 18

This suggests that **this particular P1 cell** may be a rare cardiomyocyte present in the postnatal heart that retains some immature, more proliferative characteristics generally present only in earlier embryonic cardiomyocytes. **However, further studies examining a larger number of cells with similar characteristics are needed to substantiate this observation.**

Original Text: pg 10, line 25

Taken together, single cell RNA sequencing supports the notion of heterogeneity within α MHC-expressing cardiomyocytes, particularly within the early embryonic stages, and their refinement into a more homogenous, less proliferative population by P1.

Amended Text: pg 11, line 5

Taken together, single cell RNA sequencing supports the notion of heterogeneity within α MHC-expressing cardiomyocytes, particularly within the early embryonic stages. **The lower proliferative capacity at E12.5 compared to E9.5 may be due to combined effects of reduced response to developmental growth signals and cellular migration with a shift toward processes involved in heart contraction and cellular respiration. However, congruent with the idea of heterogeneity, we were able to identify cells at E12.5 that still maintained high cell cycle activity and displayed a more “progenitor-like” gene profile.**

4. The authors should also analyze sequencing data by t-SNE instead of PCA, which notoriously fails to reveal non-linear relationships.

We appreciate the helpful suggestion. We have re-analyzed our single cell data using t-SNE algorithm, with perplexity $p = 22$ (**Rebuttal Fig 1a**). By applying k-means algorithm, we were able to identify four clusters (**Rebuttal Fig 1b**), containing the numbers of cells from each time point as follows:

	E9.5	E12.5	P1
Cluster 1 =	23	1	0
Cluster 2 =	13	7	0
Cluster 3 =	6	27	1
Cluster 4 =	1	1	42
122 cells	43	36	43

Based on this clustering, we associated Cluster 1 as enriched for E9.5, Cluster 3 for E12.5, and Cluster 4 for P1. Cluster 2 contains a mixture of E9.5 and E12.5. To further validate the clusters we have identified, we plotted different PCA components and generated a 3D scatterplot of the single cells to better understand the spatial orientation of each cell (see **Rebuttal Fig 1c** and **Rebuttal Video 1**).

Rebuttal Fig. 1. t-SNE plots with **a**, time point labeling and **b**, km-based clustering. **c**, PCA analysis comparing all three PCA components.

5. When analyzing regeneration of neonatal hearts the authors mention that they also used the Nkx2.5 driver but did not show the results.

We have now provided quantification of neonatal regeneration using the Nkx2.5 driver (**Rebuttal Fig 2e**).

a. Did the authors observe any Nkx2.5-CreER driven clones during neonatal heart regeneration?

As seen in **Rebuttal Fig 2b,e** we observed frequent Nkx2.5-CreER clones of ≥ 2 cells during neonatal heart regeneration. Specifically, within the injury and border zones, we identified a significantly greater percentage of clones of 2 cells compared to more remote regions of the heart.

b. Did the clones occur at the same or lower frequency than alphaMyHC-CreEr driven clones?

Unfortunately, with the stochastic nature of our Rainbow system, we are unable to compare the frequency of clones observed between the different strains we analyzed. As we utilized varying amounts of tamoxifen to induce minimal recombination within each of the strains, we are unable to accurately compare the absolute clone frequencies. However, the recombination events in Nkx2.5^{CreER};R26^{VT2/GK} mice resulted in clone sizes that were similar in size as those observed in α MHC^{CreER};R26^{VT2/GK}. To best represent the data, we have chosen to calculate the percentages of each clone size with respect to the total number of clones identified. However, we observed that immediately after injury clones are formed from either Nkx2.5- or by α MHC-expressing cells, suggesting that cardiomyocytes can be stimulated to undergo division in response to injury.

c. Was the clone size similar to alphaMyHC-CreEr driven clones?

Based on our quantification data (**Rebuttal Fig. 2e,f**), the clone size percentages between neonatal *Nkx2.5^{CreER}; R26^{VT2/GK}* and *αMHC^{CreER}; R26^{VT2/GK}* hearts were similar in the injury + border regions ($p = 0.81$ for single cells, $p = 0.87$ for 2 cells, and $p = 0.53$ for 2+ cells). In *Nkx2.5^{CreER}; R26^{VT2/GK}*, we observed 697 single cells, 133 clones of 2 cells, and 6 clones consisting of more than 2 cells within the injury and border zone, comprising 83.4%, 15.9%, and 0.7% of total clones, respectively. Within neonatal *αMHC^{CreER}; R26^{VT2/GK}* hearts, we counted 159 single cells, 24 clones of 2 cells, and 7 clones consisting of more than 2 cells within the injury and border zone (83.7%, 12.6%, and 3.7% of total clones, respectively).

Rebuttal Fig. 2. Neonatal mice underwent left anterior descending artery (LAD) ligation at P2 and clonal analysis performed 21 days post injury. Representative confocal images of (a) *βactin^{CreER}; R26^{VT2/GK}*, (b) *Nkx2.5^{CreER}; R26^{VT2/GK}*, and (c) *αMHC^{CreER}; R26^{VT2/GK}* hearts (left ventricle) sections. Insets show close-up of boxed regions. Quantification of cluster formation following neonatal injury indicates no significant differences between (d) *βactin^{CreER}; R26^{VT2/GK}*, (e) *Nkx2.5^{CreER}; R26^{VT2/GK}*, and (f) *αMHC^{CreER}; R26^{VT2/GK}* hearts. * $p < 0.05$ compared to remote regions.

d. What happens if the Nkx2.5 driver is activated DURING regeneration?

To specifically address this question, we performed permanent LAD ligation of P1 *Nkx2.5^{CreER}; R26^{VT2/GK}* pups and injected tamoxifen after 5 hours post-injury (we allowed 5 hours for the pups to recover from surgery). Three weeks post injury, hearts were harvested and we performed clonal analysis and quantification as previously described in our manuscript. We found that if the Nkx2.5 driver is activated **post** injury, we observe similar percentages of all clone sizes as when the Nkx2.5 driver was activated prior to injury ($p = 0.33$ for single cells, $p = 0.35$ for 2 cells, $p = 0.39$ for 2+ cells) (**Rebuttal Fig. 3**). This suggests that even if Nkx2.5 is activated in response to injury, it may not be a crucial factor in mediating regeneration, as the clone sizes are similar when Nkx2.5-expressing cells are marked before or after injury.

Rebuttal Fig. 3. Quantification of clone sizes in *Nkx2.5^{CreER};R26^{VT2/GK}* neonates that underwent permanent LAD ligation with tamoxifen administered **a**, 24 hours prior to injury and **b**, 5 hours post injury. * $p < 0.05$ compared to remote regions.

6. The author state that the 3D analysis corroborated the 2D-analysis but described an about 16-fold drop of the size of alphaMyHC-CreER driven clones between E9.5 and E12.5, which does not seem to correspond to the numbers obtained by the 2D-analysis. This should be clarified.

We apologize for the confusion – we have made the clarifications in the text in addition to formal statistical evaluation as suggested by Reviewer 2 (please see Major Comment, Point 1). In brief, we used a two-sample Kolmogorov–Smirnov distribution test to determine significance among the various strains tested for both the 2D as well as 3D analysis (**Rebuttal Fig 4**). We have also modified the cutoff ranges (black dotted lines in **Rebuttal Fig 5a, c, e, and g**) as two standard deviations away from the mean of the $\alpha\text{MHC}^{\text{CreER}};R26^{\text{VT2/GK}}$ strains. We have modified the original pie charts denoting the percentages and instead provided bar graphs denoting the number of clones identified that were above the cutoff out of the total number of clones we analyzed (**Rebuttal Fig 5b, d, f, and h**).

Original Text: pg 6, line 9

Analysis of P2 cleared hearts from $\beta\text{actin}^{\text{CreER}}; R26^{\text{VT2/GK}}$ and $Nkx2.5^{\text{CreER}}; R26^{\text{VT2/GK}}$ mice labeled at E9.5 revealed clones of similar volumes, but when labeled at E12.5, the volumes decreased by ~2-fold (Fig. 3c, g). In $\alpha\text{MHC}^{\text{CreER}};R26^{\text{VT2/GK}}$ hearts labeled at E9.5 and analyzed at P2, we observed fewer clones but of comparable volumes to the previous two strains, however, there was a dramatic ~16-fold decrease in volumes with E12.5 labeling (Fig. 3c-d, g-h). These results corroborated our 2D analysis of clone sizes and further validate the notion that while αMHC -marked cardiomyocytes retain the ability to proliferate during early embryonic development, cardiovascular progenitors are the primary contributors of cardiac growth during this time.

Amended Text: pg 6, line 9

Analysis of P2 cleared hearts from $\beta\text{actin}^{\text{CreER}}; R26^{\text{VT2/GK}}$ and $Nkx2.5^{\text{CreER}}; R26^{\text{VT2/GK}}$ mice labeled at E9.5 revealed clones of similar volumes (average $190220 \mu\text{m}^3$ and $214260 \mu\text{m}^3$, respectively), which decreased by ~2-fold when labeling occurred at E12.5 (Fig. 3c, g). In $\alpha\text{MHC}^{\text{CreER}}; R26^{\text{VT2/GK}}$ hearts labeled at E9.5 and analyzed at P2, we did not find a difference in clone volumes compared to the other two strains. However, when labeling was initiated at E12.5, significant differences were observed in volumes of αMHC -marked clones compared to $\beta\text{-actin}$ and $Nkx2.5$ ($p < 0.001$). Additionally, within αMHC -marked clones, there was a ~16-fold decrease in volumes with labeling at E12.5 when compared to E9.5. These results further validate the notion that while αMHC -marked cardiomyocytes retain the ability to proliferate during early embryonic development, cardiovascular progenitors are the primary contributors of cardiac growth during this time.

Original Text: pg 29, line 13

Student unpaired t test and one-way ANOVA were used for statistical analysis. All data are presented as mean \pm SEM. A probability value $p \leq 0.05$ was considered statistically significant. All analyses were performed with GraphPad Prism 5.04.

Amended Text: pg 29, line 18

Student unpaired t test and one-way ANOVA were used for statistical analysis. All data are presented as mean \pm SEM. Two-sample Kolmogorov–Smirnov distribution test was used to determine significance for two-dimensional clonal size analysis and three-dimensional clonal volumes analysis. A probability value $p \leq 0.05$ was considered statistically significant. All analyses were performed with GraphPad Prism 5.04.

Rebuttal Fig. 4. Two-sample Kolmogorov-Smirnov test for 2D (a,c) and 3D (b,d) clonal analysis of all three Rainbow strains labeled at E9.5 (a,b) and E12.5 (c,d).

Rebuttal Fig. 5. Quantification of clonal expansion in P2 $\beta actin^{CreER}; R26^{VT2/GK}$, $Nkx2.5^{CreER}; R26^{VT2/GK}$, and $\alpha MHC^{CreER}; R26^{VT2/GK}$ hearts labeled at E9.5 (a-d) and E12.5 (e-h). Green lines depict mean values, black dotted line depicts cut-off ranges. (b,d,f,h) Quantification of the number of clones above designated cut-off ranges.

7. To trace cardiogenic cell clones the authors used different drivers. Some of the drivers were transgenic insertions while other were knock-ins. Since the authors performed a comparative analysis, it would have been more consistent to use knock-in alleles in all cases. I do not assume that the use of transgenic insertions creates major problems but the authors should briefly mention that transgenic promoter insertions were used in some cases.

Thank you for the suggestion. As the reviewer indicated, we used transgenic as well as knock-in mice for our studies. For our clonal analysis studies, we used transgenic drivers αMHC^{CreER} (Myh11-cre/ESR1) and $\beta actin^{CreER}$ (ACTB-cre/Esr1) from the Jackson Laboratories, as well as generated our own $Nkx2.5^{CreER}$ knock-in model to examine the proliferative behavior of cardiac progenitors. Although the reviewer is absolutely correct that it is more consistent to use knock-in alleles, generating these mice would take a significant amount of time and cost. To clarify these differences, we have made the relevant changes within our manuscript as outlined below.

Original Text: pg 22, line 9

αMHC^{CreER} (Myh11-cre/ESR1), $\beta actin^{CreER}$ (ACTB-cre/Esr1) and $Wt1^{CreER}$ ($Wt1^{tm2(cre/ERT2)Wtp/J}$) were obtained from The Jackson Laboratory.

Amended Text: pg 22, line 11

αMHC^{CreER} (Myh11-cre/ESR1), $\beta actin^{CreER}$ (ACTB-cre/Esr1) and $Wt1^{CreER}$ ($Wt1^{tm2(cre/ERT2)Wtp/J}$) transgenic mice were obtained from The Jackson Laboratory.

Original Text: pg 22, line 12

Generation of $Nkx2.5^{CreER}$ Mice

Amended Text: pg 22, line 14

Generation of $Nkx2.5^{CreER}$ Knock-in Mice

Reviewer #2 (Remarks to the Author):

Major Comments:

- The results from Figure 3 are used to demonstrate the concept that more differentiated cardiomyocytes have a decreased capacity for proliferation than less differentiated progenitors. However, it also looks like fewer clusters were observed in $\alpha MHC:CreER$ analyses compared to the other strains. The $\alpha MHC:CreER$ is notoriously less efficient than the other Cre strains and the results of Sereti et al could also be the result of sampling fewer cardiomyocytes. The authors could address this concern by providing data on the total number of cardiomyocytes assessed with each Cre strain and formal statistical evaluation for a difference in the distributions of Fig 3 a, c, e, and g.

The reviewer raises a valid question and an insightful comment. We have now provided data on the total number of cardiomyocyte clones assessed from each of the CreER strain along with formal statistical evaluation for the differences in the distributions using two-sample Kolmogorov-Smirnov test. In addition, we have modified cutoffs to be greater than 2 standard deviations from the mean of the $\alpha MHC^{CreER}; R26^{VT2/GK}$ data set. These data confirm that more differentiated cardiomyocytes (marked by αMHC) have a limited proliferative capacity highlighted by marking at later time points such as at E12.5. Please refer to our response to Reviewer 1, Question 6 for further clarification.

- In their abstract, the authors claim, "Our data suggest that clonal dominance of differentiating progenitors mediates cardiac development, while a population of "immature" cardiomyocytes with a distinct molecular signature maintains homeostasis during late embryonic development and after birth." I don't think they really assessed homeostasis. I also don't think the authors demonstrated the presence of a population of "immature" cardiomyocytes in the postnatal heart. The authors should remove references to cardiac homeostasis in the text.

Thank you for the suggestion – we have now removed references to cardiac homeostasis and "immature" cardiomyocytes in the text.

Original Text: pg 2, line 17

Our data suggest that clonal dominance of differentiating progenitors mediates cardiac development, while a population of “immature” cardiomyocytes with a distinct molecular signature maintains homeostasis during late embryonic development and after birth.

Amended Text: pg 2, line 17

Our data suggest that clonal dominance of differentiating progenitors mediates cardiac development, while a subpopulation of cardiomyocytes with a distinct molecular signature may have the potential for limited proliferation during late embryonic development and shortly after birth.

3. There was one outlier “immature” cell at P1 from the single cell sequencing experiments. Not sure a whole lot can be stated based on one outlier cell.

We agree that it is difficult to make conclusions from one outlier P1 cell, however, we were curious in seeing how the transcriptional profile of this particular cell is similar or different with the other clusters of cells we observed. Nonetheless, as suggested by the reviewer, we are not drawing absolute conclusions from these findings and this data has been relocated to the supplementary section. We have also made relevant changes to the text as previously outlined in response to Reviewer 1 question 3.

4. I'd like to see the post MI adult data quantified.

We have now provided quantification data for post-MI among the different strains (**Rebuttal Fig 6**). Within the injury + border zones, we observed significant differences in the percentage of α MHC clones containing 2+ cells compared to β -actin ($p < 0.05$) as well as in $Nkx2.5$ clones of 2+ cells compared to β -actin ($p < 0.001$).

Rebuttal Fig 6. Adult mice 8 wks of age underwent left anterior descending artery (LAD) ligation clonal analysis performed 21 days post injury. Representative confocal images of (a) β -actin^{CreER}; $R26^{VT2/GK}$, (b) $Nkx2.5^{CreER}; R26^{VT2/GK}$, and (c) α MHC^{CreER}; $R26^{VT2/GK}$ hearts (left ventricle) sections. Insets show close-up of boxed regions. Quantification of cluster formation following injury indicates no significant differences between (d) β -actin^{CreER}; $R26^{VT2/GK}$, (e) $Nkx2.5^{CreER}; R26^{VT2/GK}$, and (f) α MHC^{CreER}; $R26^{VT2/GK}$ hearts. * $p < 0.05$ compared to remote regions.

5. Are the rare clusters that are observed above the numbers of clusters that would be seen by just labeling an uninjured heart with the same conditions? Lots of investigators have been pointing to multicolor lineage tracing as an approach for identifying postnatal cardiomyocyte proliferation, but this looks to be a lot harder than theorized.

To address this point, we analyzed and quantified clone formation in un-injured areas of adult hearts from each strain. When we compared the size of α MHC clones in the injury vs un-injured regions, we saw an increase in the percent of clones 2+ cells within the injury + border zones, however it was not significant ($p = 0.09$). As highlighted in our original manuscript, this suggests that myocardial injury does not induce cardiomyocyte proliferation in the adult heart (**Rebuttal Fig 7**).

Rebuttal Fig 7. Comparison of clone sizes within the **a**, injury + border zones and **b**, un-injured areas of adult β actin^{CreER}, *Nkx2.5*^{CreER} and α MHC^{CreER} Rainbow mice.

Minor Comments:

1. Figure 1 is used to develop the idea that progenitors are more proliferative than differentiating cells. However, with the labeling strategy, the patterns of clonal dominance seem to also reflect when the Cre is active. I would expect a Cre strain active during early development to result in larger clones. Since the authors used timed Cre induction to refine their argument, my comment is quite minor but probably should be addressed in the text. Thank you, as you have suggested, the timing of Cre activity is vital for our studies, as seen by the differences in clone sizes between E9.5 and E12.5 labeling. This is why we performed further studies using the tamoxifen-inducible system. To highlight this difference, we have moved a portion of the original text to a more relevant section of the text.

Original Text: pg 3, line 25

However, the use of Rainbow with a non-inducible Cre model results in high levels of recombination, and does not allow for lineage tracing at a single cell level.

Amended Text: pg 3, line 26

Single cell lineage tracing supports an essential role of cardiac progenitors in heart development

However, the use of Rainbow with a non-inducible Cre model results in high levels of recombination. To exclude the possibility that the observed single-color cell clusters could result from random recombination and expression of the same fluorescent protein within neighboring cells, we used tamoxifen-inducible Cre lines that permit tight spatiotemporal control on recombination events.

2. Can the authors confirm that the clusters labeled by β -actin::CreER experiments in Figure 3 are all clusters of cardiomyocytes and not other cell types (fibroblasts, etc)? The authors should probably address this concern in their methods section.

The reviewer is correct in that the β -actin^{CreER};R26^{VT2/GK} experiments, multiple cell types such as fibroblasts, cardiomyocytes, and endothelial cells can be labeled. We have observed clones of all these cell types in the hearts, but we focused our analysis on cardiomyocyte clones, which were consistently more abundant than others. To address the concern by the reviewer, we have stained β -actin^{CreER};R26^{VT2/GK} hearts with α -sarcomeric actinin and identified cardiomyocyte (**Rebuttal Fig 8a**) vs non-cardiomyocyte clusters (**Rebuttal Fig 8b**). We have updated the methods section of the text.

Rebuttal Fig 8. Alpha-sarcomeric actinin immunofluorescent staining of β -actin^{CreER};R26^{VT2/GK} heart sections confirming the presence of **a**, α -sarcomeric actinin+ and **b**, α -sarcomeric actinin- clones.

Original Text: pg 24, line 21

Sections were stained with WGA to allow single cell distinction (Extended Data Fig. 13). Quantification of clone number and size was performed by manual counting of the number of single color clusters and the cells within each cluster. Samples identity was hidden from the operator in order to perform quantifications in a blinded fashion.

Amended Text: pg 24, line 23

Sections were stained with WGA to identify cell boundaries (Extended Data Fig. 13). To specifically distinguish cardiomyocyte clones, sections were stained with α -sarcomeric actinin and co-localization of actinin with each clone was verified. Quantification of clone number and size was performed by manual counting of the number of single color clusters and the cells within each cluster. Samples identity was hidden from the operator in order to perform quantifications in a blinded fashion.

3. Figures S3A and S3B are not convincing for overlap of the blue clone with DDR2 or CD31. Is there a better image?

We have now provided better confocal images showing the co-localization of DDR2 and CD31 with a clone of cells (**Rebuttal Fig 9**).

Rebuttal Fig 9. Immunohistochemical staining of β actin^{CreER}; R26^{VT2/GK} clones positive for **a**, the fibroblast marker, DDR2, **b**, endothelial cell marker, CD31, **c**, smooth muscle myosin heavy chain, smMHC, and **d**, cardiomyocyte, α -sarcomeric actinin. **e**, WGA (white) was used to delineate individual cells within a clone (red pseudocolor) for counting purposes. Scale bars 100 μ m.

4. The references in the text to Figures S8B and S8C seem to be one panel off.
We apologize for this oversight - we have now updated the in-text references to the figures.

Original Text: pg. 7, line 9

Furthermore, we detected rare clusters of cardiomyocytes consisting of >2 cells derived from β actin-, Nkx2.5-, or α MHC-Rainbow labeling at P1, suggesting limited expansion of cardiomyocytes even prior to preadolescence (Extended Data Fig 8b).

Amended Text: pg 7, line 9

Furthermore, we detected rare clusters of cardiomyocytes consisting of >2 cells derived from β actin-, Nkx2.5-, or α MHC-Rainbow labeling at P1, suggesting limited expansion of cardiomyocytes even prior to preadolescence (Extended Data Fig 8c).

5. In Figure 5a, are the numbers of observed clones significantly different at the regenerate versus a more remote area?

Based on our quantification results of α MHC^{CreER}; R26^{VT2/GK} neonatal hearts following injury (**Rebuttal Fig 2f**), we did not find a significant difference in the percentages of clone sizes between the regenerate vs remote regions. However, we did observe differences in the percentages of single cells and clones of 2 cells within β actin^{CreER}; R26^{VT2/GK} and Nkx2.5^{CreER}; R26^{VT2/GK} hearts ($p < 0.05$).

Reviewers' Comments:

Reviewer #1:

Remarks to the Author:

Sereti et al made huge efforts to improve the quality of their original manuscript and to address the issues that have been raised. In addition to several new experiments, the authors also reanalyzed their data using different computational approaches and statistical methods. Although the authors could not stringently answer all my questions, mainly due to some inherent limitations of the experimental system, I am overall very satisfied with the revision.

Specific comments:

1.) During the initial review I asked whether absolute cardiac growth at E12.5 is really still driven by expansion of cardiac progenitor cells or by dividing cardiomyocytes, which might not form large clones but compensate for that "shortcoming" by their sheer number. The authors answered "However, due to the nature of tamoxifen titration to induce minimal recombination necessary for clonal analyses, we are unable to rule out the possibility that absolute cardiac growth can be driven by dividing cardiomyocytes at that time point, based on sheer number." I do see the point but I wonder why the authors -given these limitations- make the strong statement (Abstract): "...we provide direct evidence to suggest that cardiac progenitors are the main source of cardiomyocytes during murine cardiac growth." And (discussion) "Our data suggests that cardiovascular progenitors contribute to the majority of cardiac growth during embryonic development ...". I completely agree that the larger clone size of cardiac progenitor relative to cardiomyocyte clones suggest a higher contribution but since the absolute number of (small) cardiomyocyte-derived clones can not be determined the authors need to tone down this statement. In addition, I think it is mandatory to briefly discuss the limitations and consider the possibility that numerous, very small clones of cardiomyocytes might also contribute substantially to the growth of the murine heart. I am definitely not asking for additional experiments here but simply for an adequate coverage of the limitations of the model and potential alternative explanations. A brief paragraph should suffice.

2.) I agree that at this stage it is difficult to enlarge the number of analyzed cells. In fact, the authors explicitly mentioned that might have missed rare populations due to the limited number of cells analyzed, which makes the reader aware of the problem. In addition, analysis of sequenced cells was significantly improved, which strengthened the manuscript, although my original point that the authors might have missed less abundant populations due to the limited number of analyzed cells cannot be addressed by a more in-depth analysis of sequenced cells.

3.+4.) By using t-SNE instead of PCA plots the authors were able to improve comparison of different populations and pathway analysis, which is very satisfying.

5.) The authors now directly compare Nkx2.5+ and MyHC+ derived clones in the regenerating newborn mouse heart and found that the sizes were more or less equivalent, which is an interesting finding. Moreover, the apparent activation of the Nkx2.5 driver after injury is important and fascinating.

6.) The authors have eliminated the confusion about non-matching clone sizes from the 2D and 3D analysis by choosing different cut-offs and statistical methods. I have no further objections regarding this matter, although it is a bit difficult to understand (at least for a non-statistician) how precisely the different methods affected the outcome so strongly.

7.) I was certainly not asking to generate knock-in strains to replace all transgenic strains that have been used but for a short remark that sometimes transgenic strains show a less accurate reflection of endogenous gene expression patterns than knock-in strains. The authors have now mentioned in the description of the strains whether they used knock-in or transgenic mice, which

is fine. Yet, I would have preferred a halfsentence pointing out the limitations of ectopically inserted promoters.

Reviewer #2:

Remarks to the Author:

The authors addressed most of my concerns. The current version of the manuscript makes conclusions about cardiomyocyte differentiation and proliferation that are substantiated by their results. The single cell analysis seems to be more robust as well.

1. I still think the authors should use a sham control in Figure 5 for the neonatal and adult heart injuries. They use a remote, uninjured region of the heart as a control. The assumption is that a remote region of the heart undergoes cardiomyocyte proliferation at a rate similar to uninjured hearts. This is probably true for the adult heart but might not be true for the neonatal heart. The authors should acknowledge that their results can't rule out an increase in cardiomyocyte proliferative responses throughout the heart following injury. They just aren't seeing enrichment at the site of injury.
2. The authors should annotate their figures with the results of statistical testing, even if the results are not significant (Fig 3A, 3K, 3L, 5D, 5E, 5F, 5 J, 5K, 5L for example).
3. The authors will need to make sure that their figure references in the manuscript are updated (for Figure 4 especially).

We appreciate all of the helpful suggestions we have received from the reviewers. Please see below our point-by-point response to each question/concern.

Reviewer #1 (Remarks to the Author):

Sereti et al made huge efforts to improve the quality of their original manuscript and to address the issues that have been raised. In addition to several new experiments, the authors also reanalyzed their data using different computational approaches and statistical methods. Although the authors could not stringently answer all my questions, mainly due to some inherent limitations of the experimental system, I am overall very satisfied with the revision.

1. During the initial review I asked whether absolute cardiac growth at E12.5 is really still driven by expansion of cardiac progenitor cells or by dividing cardiomyocytes, which might not form large clones but compensate for that “shortcoming” by their sheer number. The authors answered “However, due to the nature of tamoxifen titration to induce minimal recombination necessary for clonal analyses, we are unable to rule out the possibility that absolute cardiac growth can be driven by dividing cardiomyocytes at that time point, based on shear number.” I do see the point but I wonder why the authors -given these limitations- make the strong statement (Abstract): “...we provide direct evidence to suggest that cardiac progenitors are the main source of cardiomyocytes during murine cardiac growth.” And (discussion) “Our data suggests that cardiovascular progenitors contribute to the majority of cardiac growth during embryonic development ...”. I completely agree that the larger clone size of cardiac progenitor relative to cardiomyocyte clones suggest a higher contribution but since the absolute number of (small) cardiomyocyte-derived clones can not be determined the authors need to tone down this statement. In addition, I think it is mandatory to briefly discuss the limitations and consider the possibility that numerous, very small clones of cardiomyocytes might also contribute substantially to the growth of the murine heart. I am definitely not asking for additional experiments here but simply for an adequate coverage of the limitations of the model and potential alternative explanations. A brief paragraph should suffice. We thank the reviewer for the clarification and apologize that we were not able to address the concern completely the first time. We have now provided a brief paragraph discussing the limitations of our model and have also toned down the statements we have made in the abstract and discussion to better fit with the limitations of the model.

Original Text: pg 2 line 10

...we provide direct evidence to suggest that cardiac progenitors are the main source of cardiomyocytes during murine cardiac growth.

Amended Text: pg 2 line 6

We provide evidence suggesting that cardiac progenitors maintain their proliferative potential and are the main source of cardiomyocytes during development....

Original Text: pg 12 line 14

Our data suggests that cardiovascular progenitors contribute to the majority of cardiac growth during embryonic development....

Amended Text: pg 11 line 24

Our study points to the possibility that cardiac progenitors are able to maintain their proliferative potential for a longer span of time and contribute to a considerable portion of cardiac growth during embryonic development.

Amended Text: pg 12 line 17

Thus, it is possible for small clones of cardiomyocytes that were not captured in the system to make sizeable contribution to murine heart growth during this time.

2. I agree that at this stage it is difficult to enlarge the number of analyzed cells. In fact, the authors explicitly mentioned that might have missed rare populations due to the limited number of cells analyzed, which makes the reader aware of the problem. In addition, analysis of sequenced cells was significantly improved, which strengthened the manuscript, although my original point that the authors might have missed less abundant

populations due to the limited number of analyzed cells cannot be addressed by a more in-depth analysis of sequenced cells.

We appreciate the reviewer's understanding about the difficulties in enlarging our sample size. Due to our awareness of this concern, we have tried to address this limitation in the manuscript and maximized the level of analysis we could perform with the number of cells we do have.

3. By using t-SNE instead of PCA plots the authors were able to improve comparison of different populations and pathway analysis, which is very satisfying.

Thank you – we appreciate your original suggestion of using t-SNE which we feel has improved our analysis.

4. The authors now directly compare Nkx2.5+ and MyHC+ derived clones in the regenerating newborn mouse heart and found that the sizes were more or less equivalent, which is an interesting finding. Moreover, the apparent activation of the Nkx2.5 driver after injury is important and fascinating.

Thank you – we did find the activation of the Nkx2.5 driver interesting as well.

5. The authors have eliminated the confusion about non-matching clone sizes from the 2D and 3D analysis by choosing different cut-offs and statistical methods. I have no further objections regarding this matter, although it is a bit difficult to understand (at least for a non-statistician) how precisely the different methods affected the outcome so strongly.

We are happy to have addressed the confusion with the 2D and 3D analysis. It was after consulting with a statistician that we were able to identify the proper method for statistical analysis in this case.

6. I was certainly not asking to generate knock-in strains to replace all transgenic strains that have been used but for a short remark that sometimes transgenic strains show a less accurate reflection of endogenous gene expression patterns than knock-in strains. The authors have now mentioned in the description of the strains whether they used knock-in or transgenic mice, which is fine. Yet, I would have preferred a half sentence pointing out the limitations of ectopically inserted promoters.

Thank you for the clarification and we apologize we were not able to completely address your initial concern. We have now provided a brief sentence regarding the limitation of ectopically inserted promoters.

Amended Text: pg 14 line 5

It should be noted that not all mouse models in this study were knock-ins, posing the potential limitations associated with use of ectopically-inserted promoters.

Reviewer #2 (Remarks to the Author):

The authors addressed most of my concerns. The current version of the manuscript makes conclusions about cardiomyocyte differentiation and proliferation that are substantiated by their results. The single cell analysis seems to be more robust as well.

1. I still think the authors should use a sham control in Figure 5 for the neonatal and adult heart injuries. They use a remote, uninjured region of the heart as a control. The assumption is that a remote region of the heart undergoes cardiomyocyte proliferation at a rate similar to uninjured hearts. This is probably true for the adult heart but might not be true for the neonatal heart. The authors should acknowledge that their results can't rule out an increase in cardiomyocyte proliferative responses throughout the heart following injury. They just aren't seeing enrichment at the site of injury.

Our preliminary experiments with sham hearts demonstrated a resemblance to what we observed with remote, uninjured regions of the heart. Nonetheless, we completely agree with the reviewer and acknowledge this in the revised manuscript that the data presented here cannot categorically rule out an increase in cardiomyocyte proliferative response throughout the heart following injury.

2. The authors should annotate their figures with the results of statistical testing, even if the results are not significant (Fig 3A, 3K, 3L, 5D, 5E, 5F, 5J, 5K, 5L for example).

Thank you for the suggestion – we have now supplied the figures with results of statistical testing.

3. The authors will need to make sure that their figure references in the manuscript are updated (for Figure 4 especially).

Thank you and we apologize for the oversight. We have now updated the figure references.